# CAAP: Context-Aware Action Planning Prompting to Solve Computer Tasks with Front-End UI Only

## Abstract

Software robots have long been used in Robotic Process Automation (RPA) to automate mundane and repetitive computer tasks. With the advent of Large Language Models (LLMs) and their advanced reasoning capabilities, these agents are now able to handle more complex or previously unseen tasks. However, LLM-based automation techniques in recent literature frequently rely on HTML source code for input or application-specific API calls for actions, limiting their applicability to specific environments. We propose an LLM-based agent that mimics human behavior in solving computer tasks. It perceives its environment solely through screenshot images, which are then converted into text for an LLM to process. By leveraging the reasoning capability of the LLM, we eliminate the need for large-scale human demonstration data typically required for model training. The agent only executes keyboard and mouse operations on Graphical User Interface (GUI), removing the need for pre-provided APIs to function. To further enhance the agent's performance in this setting, we propose a novel prompting strategy called Context-Aware Action Planning (CAAP) prompting, which enables the agent to thoroughly examine the task context from multiple perspectives. Our agent achieves an average success rate of 94.5% on MiniWoB++ and an average task score of 62.3 on WebShop, outperforming all previous studies of agents that rely solely on screen images. This method demonstrates potential for broader applications, particularly for tasks requiring coordination across multiple applications on desktops or smartphones, marking a significant advancement in the field of automation agents. Codes and models are accessible at `https://github.com/caap-agent/caap-agent`.

## 1 Introduction

Artificial intelligence (AI) agents are often commercialized as Robotic Process Automation (RPA) tools, designed to streamline business processes with minimal human intervention. Traditional RPA agents use rule-based algorithms to automate structured tasks, making them particularly effective for repetitive and standardized data processing. However, most desktop tasks are not well-suited to this rule-based approach, as they frequently involve handling unexpected situations and exceptions that are too numerous and varied to anticipate and program for in advance. In contrast, agents utilizing deep neural networks can adapt more flexibly to unforeseen situations. Studies have shown that agents trained on extensive data of expert demonstrations can effectively handle unseen tasks by imitating the decision-making styles of human experts (Humphreys et al., 2022; Shaw et al., 2023). The emergence of Large Language Models (LLMs) is now further advancing the capabilities of RPA agents. LLMs have demonstrated the potential to support complex decision-making processes with their advanced reasoning capabilities. Their In-Context Learning (ICL) ability allows them to learn meaningful actions for specific tasks with just a few examples, significantly reducing the need for large-scale data collection of expert demonstrations. As a result, research on agents leveraging ICL has been vigorously pursued in recent years (Yao et al., 2022b; Sridhar et al., 2023; Zheng et al., 2023; Shinn et al., 2024; Lin et al., 2023; Huang et al., 2022; Liu et al., 2024). As AI chips are

increasingly integrated into computers and smartphones (Intel, March 26, 2024; Samsung, January 18, 2024), and as these AI devices are poised to become the standards in the market, this new trend in the design of AI-driven RPA tools is likely to have a profound impact.

Most prior research on agents for solving computer tasks has primarily relied on HTML or the Document Object Model (DOM) as input sources, or has leveraged well-structured Application Programming Interfaces (APIs) to interact with digital environments (Humphreys et al., 2022; Liu et al., 2018; Gur et al., 2022; Kim et al., 2023; Sun et al., 2023; Furuta et al., 2023). However, the effectiveness of these methods depends on the availability of such shortcuts. They lack general applicability to broader desktop tasks that are not web-based or that require the use of applications with no provided APIs. Often, many applications and web services in the real world do not maintain rigorous development standards.

A promising approach to ensure the applicability of agents for more general computer tasks is to rely solely on visual information from the screen as input and use keyboard and mouse actions as output (Shaw et al., 2023; Cheng et al., 2024). By leveraging well-established GUI environments, such agents can be readily deployed to assist humans in performing their computer tasks. Prior research in this direction has primarily focused on using Vision-Transformers, which are a natural choice for processing screenshot images. However, even state-of-the-art Vision-Language Models (VLMs) struggle to ground actions to specific pixel coordinates (Anthropic; Zheng et al., 2024). Without a robust and reliable mechanism for processing the spatial information of GUI elements, achieving a practical agent remains unattainable.

In this paper, we propose a novel agent architecture composed of a separate visual observation module and a reasoning model (LLM), which strongly reinforces the agent's coordinate-grounding functionality. The visual observation module interprets the environment and represents it in textual form, alleviating the burden on the VLM to simultaneously process visual information and perform task-specific, reasoning-based planning. This modular approach also greatly improves the agent's adaptability a cross various applications, as each module can be trained or upgraded independently. To further support and maximize the effectiveness of our agent design, we have developed a novel prompting technique called Context-Aware Action Planning (CAAP). This technique systematically organizes the contextual information required for action-proposal and identifies syntactic structures that trigger the most effective Chain of Thought (CoT) (Wei et al., 2022). Our agent demonstrates superior task-solving performance on the MiniWoB++ and WebShop benchmark. It surpasses previous studies by utilizing only a minimal amount of human demonstration data, thereby illustrating its efficiency and effectiveness in handling complex computer tasks with limited training cost.

Our contributions are as follows:

• We present the first LLM-based agent for computer tasks that leverages ICL while emulating human behavior by using front-end User Interface (UI) channels as input-output sources. It processes input through an image-to-text model followed by an LLM, a combination that eliminates the need for extensive human labor in dataset collection before deploying the agent for new tasks. The agent operates exclusively through keyboard and mouse actions, enabling deployment on any GUI-based system without the need for special APIs to be provided.

• We introduce an effective prompting technique named CAAP, which enhances the ICL capabilities of an LLM-based agent in managing complex desktop tasks. By systematically structuring contextual information and leveraging syntactic patterns that trigger optimal CoT reasoning, CAAP significantly outperforms other agent designs.

## 2 RELATED WORK

**Agents with image inputs** Humphreys et al. (2022) introduced CC-Net, a compound model of ResNet, Transformer, and LSTM that uses image and DOM data as inputs and produces policies for keyboard-mouse

actions. It was trained via SL and RL, covering the largest number of MiniWoB++ tasks to date (104 tasks). Furuta et al. (2023) proposed WebGUM, which combines Vision Transformer (ViT) and T5 transformers. It was trained using SL to solve both MiniWoB++ and WebShop tasks, utilizing both image and HTML inputs. Shaw et al. (2023) introduced Pix2Act, the first agent to completely eliminate reliance on DOM or HTML for solving MiniWoB++ and WebShop tasks. The Pix2Act model is built upon the Pix2Struct architecture (Lee et al., 2023) and uses only image inputs. While these studies provide valuable insights and serve as important benchmarks, their methods are impractical for real-world applications as they require around a million or more human demonstration data samples, demanding thousands of hours of human labor for training. Cheng et al. (2024) introduced SeeClick, which is aided by the strong reasoning capabilities of the VLM. SeeClick required a much smaller dataset of 2.8K human demonstrations to train for MiniWoB++ tasks. It fell short in overall performance, however, achieving less than a 70% success rate on the 55 tasks it covered. A common feature of all these approaches is the integration of image interpretation and action planning models within a single architecture. This rigid design poses a significant limitation: when new UI elements need to be recognized or additional actions supported for a new task, the entire architecture must be retrained.

**LLM-based agents with ICL**    To reduce dependence on large quantities of human demonstration data, several studies have explored the use of LLMs for action planning. Yao et al. (2022b) developed the ReAct model, which evaluates the current state before determining subsequent actions. Kim et al. (2023) introduced the Recursive Criticism and Improvement (RCI) agent, which enhances action planning through iterative self-critique, highlighting the importance of advanced reasoning techniques in agents. Huang et al. (2022) proposed Inner Monologue (IM), a mechanism that integrates real-time environmental states into action planning via an explicit feedback loop. Inspired by IM, Sun et al. (2023) introduced AdaPlanner, which prompts LLMs to generate action plans as Python functions and refine them based on feedback from execution failures. Despite their innovations, all these approaches share a common limitation: their architectures are designed to operate solely with HTML inputs, restricting their applicability to constrained, well-curated web environments.

## 3    MODULARIZED GUI AGENT FOR COMPUTER TASKS

Our agent is composed of three distinct modules: the visual observer, which processes and interprets image inputs; the action proposer, which suggests subsequent actions; and the action executor, which carries out the proposed actions. The agent operates by iteratively repeating a cycle in which these modules execute sequentially. Each module is further divided into sub-modules based on its functionality, as illustrated in Figure 1. This modular architecture provides substantial benefits over monolithic agent designs, such as those utilizing VLMs, particularly in flexibility, resource efficiency, and scalability. Modular designs enable targeted updates to specific components, ensuring more robust and stable improvements without impacting the entire system. Additionally, they reduce computational complexity during training. When needed, functional sub-modules can be seamlessly integrated alongside existing ones to further enhance system performance.

### 3.1    VISUAL OBSERVER[1]

The visual observer module employs a two-step approach to transform visual information from a screen into the linguistic domain. Initially, in the detection stage, UI elements are identified, and their spatial locations are extracted. Subsequently, these elements undergo further analysis through advanced object understanding techniques, during which detailed attributes of each element are captured. This processing results in the information being organized into structured text, ready to be handed to the action proposer as the representation of the current state.

---

[1]While VLMs may eventually advance enough to replace the visual observer in the future, it would still serve as a valuable complementary tool for interpreting screenshots. For instance, the action proposer's LLM model could be

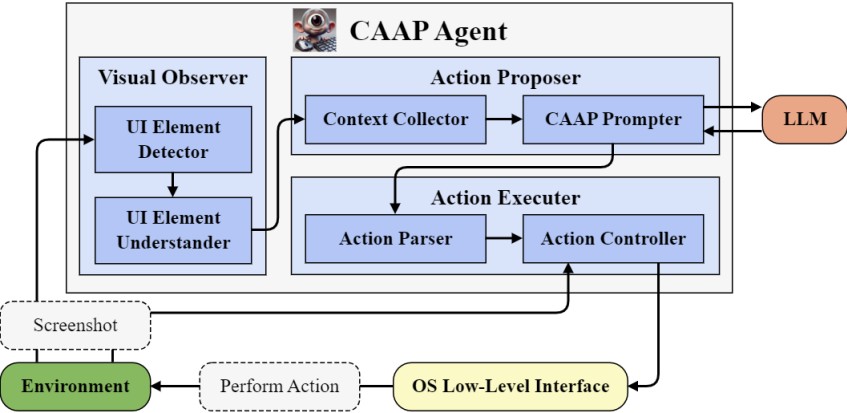

Figure 1: The architecture and task-solving flow of the CAAP agent. The agent interprets a screenshot captured in the computer environment through the visual observer. The action proposer leverages the reasoning capabilities of the LLM to determine the next actions to take based on the observed state. Once actions are decided, the action executer applies the corresponding keyboard and mouse actions to the environment via the OS interface. This sequence of processes across the three modules continues until the task is completed.

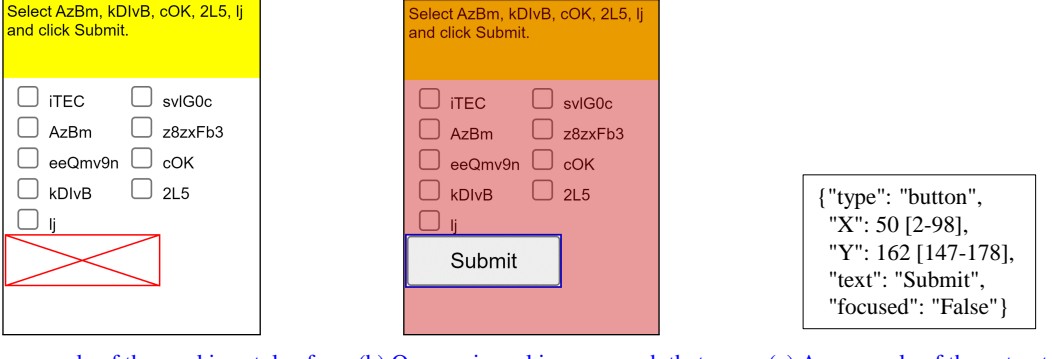

(a) An example of the masking style of the original Pix2Struct that covers the target element with an X-box.

(b) Our semi-masking approach that highlights the target element by outlining it and darkening its surrounding.

(c) An example of the extracted features from the image of (b) by our visual observer.

Figure 2: Comparison of the masking methods for the original Pix2Struct and our UI element understanding model, and an example of the extracted features for our vision observer. While the original Pix2Struct outputs text in a HTML-like format, our model is finetuned to return JSON-style text.

**UI element detection**   For the detection and extraction of location information of all UI elements present in screenshots, we fine-tuned the YOLO model. The YOLO model, known for its rapid and precise detection in one shot, typically extracts object type information along with bounding boxes. However, we utilize only the bounding boxes, leaving the extraction of type information to the subsequent UI element understanding module for greater accuracy.

---

replaced with a VLM capable of processing both the screenshot image and the textual representation provided by the visual observer, achieving a synergistic integration of multimodal capabilities.

**UI element understanding**   We use the Pix2Struct model (Lee et al., 2023) to extract detailed attributes from all UI elements identified during the detection stage. We fine-tune the original Pix2Struct model to match the specific output format required for our task. During this fine-tuning process, we employ a different masking technique from Pix2Struct style, as depicted in Figure 2 (a). Our masking technique highlights the target UI element by outlining its region and darkening the surrounding image areas, as demonstrated in Figure 2 (b). This method preserves the pixel-level details of the targeted element while helping the model learn the spatial relationships between the element and its adjacent elements. The structured output from the visual observer is presented as shown in Figure 2 (c), with the range of extractable attributes for each element detailed in Table 4 of Appendix A.

### 3.2   ACTION PROPOSER

Upon receiving the representation of the current screen, the action proposer is employed to determine possible actions for completing the task. The proposal process is facilitated by its two key components: a context collector and a CAAP prompter. The context collector gathers vital contextual information that aids in task resolution. This includes exemplary demonstrations, the ultimate task goal, the history of actions performed so far, and the range of possible actions that the language model can choose from, all formatted as text. The CAAP prompter then uses this information to construct prompts that are sent to the language model, eliciting responses that guide the next steps. The CAAP prompting method, one of our major contributions in this paper, is discussed in detail in Section 4.

### 3.3   ACTION EXECUTOR

The actions determined by the action proposer are implemented through a low-level interface to effect changes in the environment by the last module, action executor. It first identifies a series of recommended actions by parsing the response text from the action proposer. Then, it executes these actions using the keyboard and mouse interfaces of the operating system. After each action, it receives an updated screenshot, halting further actions if a change in the environment is detected. The types of actions that can be executed include mouse actions such as clicking, pointing, and scrolling, as well as keyboard actions like typing and using common shortcuts. These are enumerated in Table 5 of Appendix B and implemented as Python functions using the PyAutoGUI package.

## 4   CAAP PROMPTING

The performance of LLM-based agents is significantly influenced by the structure of their prompts. Our prompting approach is specifically designed to enable the LLM to effectively utilize surrounding context information essential for decision-making. This approach aims to elicit a CoT reasoning process. The strength of the CoT mechanism lies in its ability to connect disparate pieces of context information, progressively reinforcing their cohesion and enriching the reasoning process. CAAP prompting is specifically developed to enable the effective utilization of this mechanism during the action proposal phase. To achieve this, it actively incorporates both explicit and implicit strategies to guide the LLM's CoT reasoning. The design of CAAP prompting is systematically structured as illustrated in Figure 3.

### 4.1   EXPLICIT AND IMPLICIT CoT INDUCEMENT

**Explicit inducement of CoT**   Inspired by human action planning, we identified four essential types of surrounding context information necessary for determining subsequent actions. These contextual elements are conveyed to the LLM along with CoT-inducing instructions, which explicitly guide the model to establish semantic connections between them. Using CoT-inducing instructions, the LLM examines the actions taken so

```
┌─────────────────────────────────────────────────────┐
│ Human demonstrations                                │
│    • Exemplars of action trajectory with rationale  │
│                                                     │
│ Surrounding context information                     │
│    • Task description                               │
│    • Action trajectory                              │
│    • Visual observation description                 │
│    • Candidates of action types                     │
│                                                     │
│ CoT-inducing instructions                           │
│    • Instruction for reviewing action trajectory    │
│    • Instruction for reviewing visual observation   │
│    • Instruction for improving action plan          │
│    • Instruction for deciding next actions          │
│                                                     │
│ Extra guidelines                                    │
└─────────────────────────────────────────────────────┘
```

Figure 3: Content design for the CAAP prompting.

far to uncover the relationship between the task objective and the action trajectory. It then infers the contextual meaning of these actions and the updated state of the visual environment. Subsequently, referencing the candidates of action types, it refines the subsequent action plan considering the current situation and outputs actionable guidance in a structured text format.

**Implicit inducement of CoT**    In addition to CoT-inducing instructions, we enhance the implicit induction of chain-of-thought (CoT) reasoning in large language models (LLMs) through human demonstrations. By providing exemplars of few-shot task-solving action trajectories from experts, LLMs can be guided to reason step-by-step Wei et al. (2022). However, an over-reliance on demonstrations can lead LLMs to rigidly mimic the examples, making them less capable of adapting to exceptional scenarios. To address this, we augment each action in the demonstrations with corresponding rationales, enabling the LLM to better comprehend the contextual information relevant to the action. These rationales contain detailed natural language descriptions of the actions and the associated circumstances. The methodology for generating these rationales using LLMs is described in the 'human demo dataset' section of Appendix C.

The prompts both explicitly and implicitly inducing CoT reasoning and the subsequent inference processes of the LLM can be found in Appendix D, specifically in Figures 6 and 7, respectively.

## 4.2    CAAP COMPONENTS

**Human demonstrations**    Providing few-shot exemplars within the prompt is a common practice that enhances the advanced reasoning capabilities of LLMs. CAAP prompt begins with few-shot human demonstrations, which are sequences of actions with rationales for solving tasks similar to the given objective.

**Surrounding context information**    This part contains all the essential information for the language model to propose the next actions. It includes a description of the target task, a record of past actions, state descriptions from the visual observer, and the available range of actions for the agent.

**CoT-inducing instructions**    The section on CoT-inducing instructions contains phrases that encourage the LLM to link the surrounding context information, thereby inducing CoT. *Instruction for reviewing action trajectory* encourages the agent to review its actions taken so far. *Instruction for reviewing visual observation* prompts the agent to examine the current state's observations. *Instruction for improving action plan* requests

the formulation of a subsequent action plan, taking into account the self-generated text thus far. Finally, the *Instruction for deciding next actions* provides guidance on the desired response format and requests a list of actions to be taken in the subsequent steps.

**Extra guidelines**    The final section of the CAAP prompt primarily features phrases designed to safeguard the LLM's output. It incorporates additional directives specifically tailored to reduce common errors made by the particular LLM in use.

## 5    EXPERIMENTS

### 5.1    MINIWOB++

MiniWoB++ is a highly suitable benchmark for evaluating an agent's ability to handle a wide range of scenarios encountered during computer task execution, as it encompasses over a hundred of fundamental tasks (Liu et al., 2018; Shi et al., 2017; Min). As an extension of OpenAI MiniWoB, this benchmark includes a diverse spectrum of tasks, ranging from simple activities such as button clicks and form filling to more complex tasks like email forwarding and flight booking. We utilize the MiniWoB++ benchmark to thoroughly assess our agent's effectiveness in navigating and solving diverse tasks within a simulated computer environment.

Previous studies using the MiniWoB++ benchmark often choose subsets of tasks most suitable for demonstrating their methods. We have also selected a total of 73 MiniWoB++ tasks that meet the following criteria: (1) tasks that consist of UI elements of types recognizable by our visual observer module, as listed in Table 4 of Appendix A; (2) tasks where none of UI elements extend beyond the default screen area of 160x210 pixels; and (3) tasks that do not require color recognition, as our visual observer module is not yet trained for this capability.

For each selected task, we run at least 50 repetitions with different seeds to estimate the Success Rate (SR) for that task. Across all task types, we consistently use the identical CAAP prompt template without making any manual adjustments for specific tasks, unlike some other works (Kim et al., 2023).

**Models and datasets**    YOLO and Pix2Struct models within the visual observer have been fine-tuned to enable accurate recognition of images presented in MiniWoB++ environment. The image dataset for training has been collected through a process of capturing screenshots and annotating the UI elements within. This dataset consists of 1.76K screenshot images, gathered and annotated over a span of 10 hours by two individuals.

For human expert demonstrations, we generate between 0 and 5 demonstrations per task, tailored to the task's complexity, amassing a total of 97 demonstrations for our experiments. We record the trajectory of each action performed by the demonstrators, alongside screenshots taken at the time of each action. These screenshots are subsequently analyzed to extract UI elements, enabling the identification of the UI component interacted with during mouse-related actions, such as clicks. This demonstration data have been collected by a single individual over a span of approximately two hours.

For the action-proposal language model, we utilize the Azure OpenAI gpt-4-0125 model. The Azure OpenAI API features the 'function calling' capability, thus our prompt does not directly include the list of action types (as depicted in Figure 3). Instead, this contextual information is conveyed to the language model through arguments in the function calling feature. An example of how the function calling is applied is illustrated in Figure 8 in Appendix E.

**Experiment result**    Table 1 compares the performance of our approach with other studies reported on the MiniWoB++ benchmark. Since each study reports results over different sets of tasks (See Figure 9 of

Table 1: Comparison with other image-only methods for solving MiniWoB++ problems.

| Method | Modality | Image datasize | Demo datasize | Reported SR | Reported tasks |
|---|---|---|---|---|---|
| Human | Image | - | - | 93.5% | 104 |
| CC-Net (no DOM) | Image | 0 | 2.4M | 24.0% | 104 |
| Pix2Act | Image | 0 | 1.3M | 96.2% | 59 |
| SeeClick | Image | 0 | 2.8K | 69.4% | 55 |
| CAAP | Image | 1.8K | 0.1K | 94.5% | 73 |

Table 2: Comparison with other image-only methods for solving WebShop problems. The LLM model used for the action proposer in the CAAP method is specified in parentheses.

| Method | Modality | Image datasize | Demo datasize | Task score |
|---|---|---|---|---|
| Human | Image | - | - | 82.1 |
| Pix2Act | Image | 0 | 1K | 46.7 |
| CAAP (OpenAI gpt-4-0125) | Image | 0.3K | 1 | 62.3 |
| CAAP (Claude 3.5 sonnet) | Image | 0.3K | 1 | 70.01[3] |

Appendix F.), we have included both the SRs reported in the literature and the number of tasks used in each case. First, CC-Net, which covers the largest set of tasks, was trained on 2.4M human demonstration data across 104 tasks. However, when using only image input (excluding DOM), its performance dropped to 24%, indirectly highlighting the challenge of solving computer tasks using image inputs alone. Pix2Act, trained on 1.3M human demonstrations across 59 tasks, achieved an SR of 96.2%, demonstrating the feasibility of agents solving computer tasks with image input only. SeeClick, despite using a relatively small dataset of 2.8K demonstrations to learn click locations on screenshots, achieved an SR of only 69.4% across 55 tasks, showing limited effectiveness. In contrast, our approach, using only 97 demonstration samples, achieved a high task SR of 94.5% across 73 tasks. Although an additional 1.76K samples[2] were required to train the model for image input interpretation due to the modular nature of our approach, our method still demonstrated its effectiveness on a broader range of computer tasks with about 1.86K samples in total. Task-level SRs are detailed in Table 7 of Appendix G.

Of the 73 tasks included in our experiments, six tasks—*drag-circle*, *drag-single-shape*, *find-greatest*, *generate-number*, *odd-or-even*, and *sign-agreement*— were excluded during the generation of training data for the visual observer, making them unseen tasks for our agent. Despite this, the agent successfully completed these tasks with an average success rate of 95.3%. These results underscore the benefits of our LLM-based approach compared to the prior end-to-end learning method that relied on imitation learning from human demonstrations. As long as the new tasks do not introduce completely unfamiliar types of UI elements that the visual observer cannot recognize, our agent can effectively manage these unseen tasks by utilizing the knowledge and reasoning capabilities of the LLM.

## 5.2 WEBSHOP

WebShop (Yao et al., 2022a) is a simulated e-commerce web environment well-suited for assessing an agent's ability to handle real-world scenarios. To successfully complete WebShop tasks, the agent must understand textual instructions and requirements for the target product and navigate various types of web pages to locate

---

[2]These image samples were semi-automatically annotated using a custom tool shown in Figure 10 of Appendix H.

[3]This score is measured on 100 out of 500 test instructions, and the final score is subject to further updates.

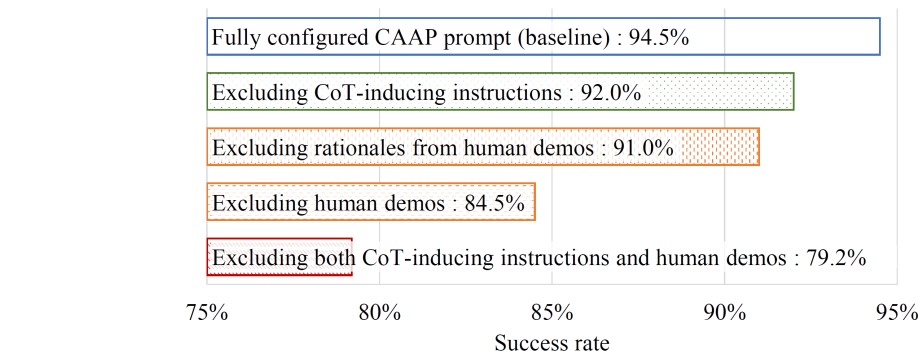

Figure 4: The effects of different CAAP components on MiniWoB++ task performance.

it, out of a pool of 1.18 million real products scraped from amazon.com. The environment provides a total of 12,087 instructions, divided into 10,587 for training, 1,000 for development, and 500 for testing. Among them, we utilized only the 500 test instructions for evaluation purposes.

**Models and datasets** The web browser for the WebShop environment can be resized freely, and for convenience, we used a fixed resolution of 1276 × 1153 pixels for both data collection and evaluation. Unlike MiniWoB++, the dataset used to train the visual observer for WebShop was generated entirely through an automated process. Given an instruction, a simple helper agent we created extracted a search term, performed a search, and clicked the first item in the results. Screen images were captured throughout this process, and annotation data was collected using Selenium for each image, resulting in a total of 300 annotated samples. The action-proposal model, supported action types, and function-calling approach are entirely consistent with those employed in the MiniWoB++ experiments. For the evaluation of the CAAP agent, a single human demonstration is applied universally to all instructions.

**Experiment result** There has been limited research on constructing agents that rely solely on image inputs, and Pix2Act is the only method whose performance has been validated on the WebShop benchmark. Table 2 presents a comparison between our approach and Pix2Act on WebShop. (Comparisons with other agent-based methods that rely on additional HTML input, on top of image input, are presented in Appendix I.2.) While Pix2Act employed 1,012 human demonstrations to train its agent model, our approach relied on just a single human demonstration, along with 300 samples collected through a fully automated process for training the visual observer module. Despite the significantly reduced manual data collection effort, our agent achieved a task score of 62.3, a meaningful improvement over Pix2Act's score of 46.7, with a margin of 15.6. Although this result does not yet reach human-level performance, it demonstrates the potential of our approach to solve real-world computer tasks with minimal human effort in data collection.

### 5.3 ABLATION STUDIES

**Impact of CoT inducement on task-solving performance** Figure 4 presents experimental results illustrating the impact of CoT inducement on task performance. The results show that incorporating CoT-inducing instructions leads to a significant improvement in performance. Specifically, as shown in the figure, adding CoT-inducing instructions increases task SRs by 5.3% (from 79.2% to 84.5%) in the absence of human demonstrations and by 2.5% (from 92.0% to 94.5%) when human demonstrations are included. Moreover, augmenting human demonstrations with rationales further enhances task SRs by 3.5%, raising them from 91.0% to 94.5%. Additionally, removing human demonstrations entirely results in a 10.0% drop in task

Table 3: Performance comparison of the CAAP and RCI promptings. The *RCI agent with CAAP* is configured by replacing the prompting method of the original RCI agent with the CAAP approach.

| | | RCI agent with CAAP | RCI agent (original) |
|---|---|---|---|
| Average SR | | 0.840 | 0.769 |

SRs, while removing CoT-inducing instructions in addition leads to a substantial decrease of 15.3%. These findings highlight that CAAP prompting, by explicitly and implicitly guiding the LLM to think step-by-step and establish contextual relationships before proposing actions, significantly improves task SRs in solving computer-based tasks. Detailed per-task results are presented in Table 11 of Appendix J.

**Comparison of the CAAP and the RCI prompting**   Since our main experiments combine GUI-based perception and execution with the CAAP prompting technique, we conducted additional experiments to isolate the effect of CAAP prompting as part of an ablation study. Without modifying the input/output channels, we tested how the performance of the RCI agent (Kim et al., 2023) improves when its prompting mechanism is replaced by our CAAP prompting. We first executed the original RCI code from the authors' official GitHub repository without modifications, testing only the 47 tasks listed in the repository by 50 times and taking the average.[4] Then, we replaced the prompting method with CAAP and reran the tests. To ensure a fair comparison, we used identical seeds for task instances, the same demo set provided by the repository, and the same LLM for queries.

As described in Table 3, the results demonstrate that applying CAAP prompting to the RCI agent significantly increased the average task-solving SR by 7.1%, from 76.9% to 84.0%. (Task-level results can be seen in Table 12 of Appendix K.) The prompting methods of RCI and CAAP exhibit two key differences. First, the RCI approach tends to modify even correct action plans during the iterative critique and revision process. In contrast, the CAAP method, by focusing on thorough context review rather than critique, is able to avoid this issue when formulating action plans. Second, RCI executes actions sequentially based on a fixed initial plan, failing to adapt to changes in the environment that occur during execution. Conversely, the CAAP method is designed to account for state changes resulting from prior actions. This experiment underscores the effectiveness of our CAAP prompting, confirming its potential to enhance the performance of LLMs in computer tasks.

## 6 CONCLUSION

In this paper, we introduce a modularly designed GUI agent model that emulates the way humans solve tasks on computers. Our modular design not only effectively enhances pixel coordinate detection—a critical challenge faced by integrated VLM architectures—but also addresses the limitations of existing models that rely heavily on imitation learning with large amounts of human demonstrations. Furthermore, we propose the CAAP prompting mechanism to improve the action-proposal capabilities of LLM-based agents for computer tasks by guiding the LLM to fully utilize the CoT reasoning mechanism. Evaluations on the MiniWoB++ and WebShop benchmarks demonstrate that our agent model achieves superior performance compared to the previous methods while requiring significantly less data. By adopting a modular approach that separates visual perception, action proposal, and action execution, our method allows for easy adaptation to new tasks or model upgrades. Additionally, our agent is designed to operate beyond web environments and across inter-application settings, offering broader applicability and significant advantages in terms of generality.

---

[4]The success rate of the RCI agent we observed deviates from the results reported in the literature, despite our diligent efforts to replicate the experimental setup exactly. Upon inquiry, the RCI authors indicated that they no longer had access to the complete experimental logs used in their publication.

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

# APPENDIX

## A ELEMENT COVERAGE OF CAAP AGENT

Table 4 provides a comprehensive overview of the types of front-end elements handled by the agent, predominantly including well-known types of front-end objects. Beyond the traditional categories, this study introduces the concept of 'tabled text' to describe texts commonly embedded within table structures, and presents 'draggable text' to represent text elements of which its text can be partially selected. These additions ensure extensive coverage of both interactive and static elements found in varying types of applications, thereby enhancing the understanding capability of the visual environmental states. Additionally, our agent covers conventional shapes and icons. The shape type includes circle, triangle, and rectangle, distinguished by their subtype values. Moreover, the icon type encompasses subtypes such as "back," "delete," "important," "forward," "reply," "search," and "send," as identified in MiniWob++.

Table 4: The front-end element coverage of CAAP agent.

| Type | Subtype | Checked / focused / highlighted | Text | Coordinates |
|---|---|---|---|---|
| Text | - | √ | √ | √ |
| Hyperlink | - | √ | √ | √ |
| Button | - | √ | √ | √ |
| Radio | - | √ | √ | √ |
| Checkbox | - | √ | √ | √ |
| Dropdown | - | √ | √ | √ |
| Input field | - | √ | √ | √ |
| Text area | - | √ | - | √ |
| Resize handle | - | - | - | √ |
| Scrollbar | - | - | - | √ |
| Tabled text | - | √ | √ | √ |
| Draggable text | - | - | √ | √ |
| Shape | √ | - | - | √ |
| Icon | √ | √ | - | √ |
| Image | - | - | - | √ |

## B    ACTION COVERAGE OF CAAP AGENT

Tables 5 and Table 6 detail the action types supported by the CAAP agent and the requisite parameters, respectively. The action executor operates using atom actions based on mouse and keyboard interactions, specifically: mouse down, mouse move, mouse up, and type a key. However, when conveying candidate actions to an LLM, it is more effective to transmit them as high-level actions capable of altering the environment. Consequently, as indicated in Table 5, actions have been delineated into units such as clicking on an element, clicking an empty position, clicking on an element during pressing control key, and typing text.

Table 5: The action coverage of our agent. 'Element ID' is the ID number of the screen element to act on. The number is designated during the element detection procedure. 'Text to Type' is which text to type on. 'Coordinates' are the X and Y coordinates of the mouse down location.

| Name | Required | Description |
|---|---|---|
| click_element | element_id | This action moves the mouse pointer to a screen element and performs a left-click to activate that element. |
| click_new_point | x, y | This action moves the mouse pointer to a screen location and performs a left-click on the location. |
| control_click_element | element_id | This action click on a screen element while holding down the 'control' modifier key. It is used to select multiple elements. |
| type_text | string_to_type | This action makes keyboard typing actions to enter the text into a specific screen element, Clicking on the element is essential to give it focus, without which typing the text is impossible. |
| point_element | element_id | This action moves the mouse pointer to on top of an UI element without clicking it. This sometimes activates the element and reveals hidden menus or scrollbar. |
| press_control_A | - | This is 'Select All'. All text in the activated text field is highlighted. |
| press_control_C | - | This is 'Copy'. Highlighted text is copied to the clipboard. |
| press_control_V | - | This is 'Paste'. Text stored in the clipboard is pasted into the selected area. |
| drag_mouse_hold_down | x, y | This action initiates the drag action sequence by clicking and holding down the left mouse button. It is used to move objects on the screenor to highlight a block of text. This action marks the starting point of the dragging move. |
| drag_mouse_move | x, y | This action is the middle of the drag action sequence. It moves the mouse pointer while holding down the left mouse button. When used to move an object on the screen, the object will be dragged to the current mouse location. When used to highlight text, it will highlight up to the current mouse location. |
| drag_mouse_release | - | This action marks the end of the drag action sequence. It releases the left mouse button, indicating that the drag move is finished. starting point of the dragging move. |

Table 6: Parameter candidates listed in the 'required' column of Table 5.

| Name | Type | Description |
|---|---|---|
| element_id | integer | The id number (ranged from 1 to N) of the screen element to point at. A list of elements will be provided with the corresponding id numbers. |
| x | integer | The x coordinate of the click location. |
| y | integer | The y coordinate of the click location. |
| string_to_type | string | The text to type. |

## C    MINIWOB++ EXPERIMENT DETAILS

**Models for visual observer**    We utilize Ultralytics YOLOv8.0.143 (ult), employing the default hyper-parameters with the exception of 'box', 'cls', 'dfl', and 'max_patches' set to 0.9, 0.05, 0.05, and 512, respectively. For Pix2Struct, we use the base model provided at `https://huggingface.co/google/pix2struct-base`, without any modifications to the default configuration.

**Fine-tuning dataset for visual observer**    Data collection for fine-tuning the visual observer involves a two-step process: screenshot collection and annotation of the collected UI elements. Initially, the target task is manually performed while screenshots are taken at each step. Image comparisons are then conducted to remove duplicate screenshots. From 10 episodes for each of the 73 tasks, a total of 1,768 screenshots were collected. In the second stage, a GUI-based annotation tool, as depicted in Figure 10 of Appendix H, is utilized. Each UI element within the screenshots is marked with bounding boxes, and specific attributes are assigned. These attributes are subsequently stored in a JSON format. Data augmentation techniques are also employed to enhance the dataset. These techniques include color adjustments, minor scaling of bounding box sizes, and the random insertion of Gaussian noise. As a result, the total number of annotated data items is tripled compared to the original dataset.

The YOLOv8 model is fine-tuned using the screenshots and their bounding boxes around the UI elements. The Pix2Struct model is fine-tuned using the masked screenshots (Figure 2(a)) and the corresponding element attributes, excluding the positions of bounding boxes.

**Human demo dataset**    To further enrich the dataset, we augmented each action with a rationale statement. As illustrated in Figure 4, the performance of the CAAP agent significantly improves when rationales for the demonstrated actions are also provided. Instead of relying on demonstrators to articulate their reasoning—a method that is impractical and unscalable for deploying our CAAP agent to a wider range of applications—we employed an LLM (*i.e.* GPT-4) to generate these rationales in a fully automated way. For each action, a prompt comprising both the visual state and the corresponding action was presented to the LLM, as detailed in Figure 5. The rationales were then integrated with the action trajectories to produce comprehensive, high-quality demonstrations. An example is shown in the 'Expert Demonstration' section of the prompt in Figure 6.

**Train and test episode split**    To differentiate between the train datasets and the test dataset, we assign distinct seed numbers when generating individual task instances within MiniWoB++. Image annotation data was collected from tasks with seed numbers ranging from 1000 to 2999, while demonstration data was sourced from seeds ranging from 3000 to 3999. Testing was conducted using seeds ranging from 0 to 999.

# D PROMPT EXAMPLES

---

Tasks can be completed by applying appropriate actions in sequence.

Below is a record of a successful completion of the given task, demonstrated by an expert.
(Note: The trainee has added the "reason" part for each action in the record, but it may not accurately describe the reasoning used by the expert who performed the task. Do not assume that the written reason is correct.)

TASK:
Select EiTE and click Submit.

Action History:
action_1: {name: start, reason: "Initiating the task."}
**action_2**: {name: click_element, arg: {type: radio, X: 31 [10-52], Y: 62 [55-70], checked: False, text: "EiTE"}}
action_3: {name: click_element, arg: {type: button, X: 50 [2-98], Y: 114 [99-130], text: "Submit", focused: False}}

We want to explain to a trainee why the action_2 was made.

Before the action_2, the status of the computer screen was as the following:
(Note: Coordinates are given in the form: center_x [left_edge_x-right_edge_x], center_y [top_edge_y-bottm_edge_y])
demo_element_1: {type: radio, X: 31 [10-52], Y: 62 [55-70], checked: **False**, text: "EiTE", visible: True}
demo_element_2: {type: radio, X: 38 [10-67], Y: 80 [73-88], checked: False, text: "vAzBm9", visible: True}
demo_element_3: {type: button, X: 50 [2-98], Y: 114 [99-130], text: "Submit", focused: False, visible: True}

After the action_2, the status of the computer screen was as the following:
demo_element_1: {type: radio, X: 31 [10-52], Y: 62 [55-70], checked: **True**, text: "EiTE", visible: True}
demo_element_2: {type: radio, X: 38 [10-67], Y: 80 [73-88], checked: False, text: "vAzBm9", visible: True}
demo_element_3: {type: button, X: 50 [2-98], Y: 114 [99-130], text: "Submit", focused: False, visible: True}

First, explain how the action_2 (both its type and its arguments) was chosen by the expert, and why it was necessary.
Since the trainee cannot view the screen, always provide a detailed description as specified whenever you refer to a screen component in your response.
Second, describe what happened after the action, as shown on the screen.

Answer in one paragraph.

---

Figure 5: A prompt example of generating a rationale for an agent action.

Tasks can be completed by applying appropriate actions in sequence.

### Expert Demonstrations ###
For example, given below are the demos showing the correct sequence of actions for each corresponding task:
DEMO_1 = {
TASK:
Select Janella from the list and click Submit.

Action History:
demo_action_1: {name: start, reason: "Initiating the task."}
demo_action_2: {name: click_element, arg: {type: dropdown, X: 76 [0-151], Y: 66 [56-76], text: "Storm"}, reason: "The expert chose action_2, which is a click_element action, to interact with the dropdown menu on the screen. The dropdown menu, identified as demo_element_1, was located at the center of the screen with coordinates X: 76 [0-151] and Y: 66 [56-76], and had the text "Storm" visible on it. Clicking on this dropdown was necessary to expand the list of options and select the desired item, "Janella," from the list. The dropdown needed to be expanded because the task required selecting a specific item that was not currently visible or selectable on the screen. After the action was performed, the dropdown menu remained visible, but the "Submit" button (demo_element_2) became temporarily invisible, indicating that the dropdown list expanded over it. New elements appeared on the screen, which were the options within the dropdown menu, including "Selie," "Janella," "Storm" (now highlighted to indicate it was the previously selected option), "Gena," "Betti," and "Chrissie," with their respective coordinates and visibility status. This expansion allowed the expert to proceed with selecting "Janella" from the list."}
demo_action_3: {name: click_element, arg: {type: tabled_text, X: 76 [0-151], Y: 117 [109-125], text: "Janella"}, reason: "The expert performed action_3, which is a click_element action, to select "Janella" from the dropdown list. This action was necessary because the task required selecting a specific name, "Janella," from the list of options that appeared after expanding the dropdown menu. The expert identified the correct element to click by its type (tabled_text), its coordinates (X: 76 [0-151], Y: 117 [109-125]), and its text content ("Janella"), which was visible on the screen among other options. After clicking on "Janella," the dropdown list collapsed, making the "Submit" button (demo_element_2) visible again, and the dropdown menu now displayed "Janella" as the selected item. The other names in the list ("Selie," "Storm," "Gena," "Betti," and "Chrissie") were no longer visible, indicating that the dropdown had retracted and the task could proceed to the next step, which was to click the "Submit" button."}
demo_action_4: {name: click_element, arg: {type: button, X: 50 [1-98], Y: 96 [79-112], text: "Submit"}, reason: "The expert chose action_4, which is a click_element action, to interact with the "Submit" button on the screen. This action was necessary to complete the task of selecting "Janella" from the dropdown list and then submitting the choice. The "Submit" button was identified by its type (button), its coordinates (X: 50 [1-98], Y: 96 [79-112]), and its text content ("Submit"), which was visible and accessible on the screen. Clicking the "Submit" button was the final step required to confirm the selection of "Janella" and proceed with the task. After the action was performed, the screen status changed to an Empty Screen, indicating that the task was completed successfully and the application or window that was previously open had closed or moved to the next stage, leaving no elements visible on the screen."}
}

We are solving a similar task.
You are given the history of actions made correctly by the user so far, and current screen status which is the result of those actions.

### Task Description ###
Select Helli from the list and click Submit.

### Action History ###
action_1: {name: start}

### The UI Element List of the Current Screen That You Can Take Next Actions On ###
(Note: Coordinates are given in the form: center_x [left_edge_x-right_edge_x], center_y [top_edge_y-bottm_edge_y])
- ID: element_1, data: {type: dropdown, X: 76 [1-152], Y: 66 [55-76], text: "Theodora", focused: False}
- ID: element_2, data: {type: button, X: 50 [1-98], Y: 96 [80-112], text: "Submit", focused: False}

### Instructions ###
What should be the next actions(action_2, action_3, ...) that can be performed on the current screen?
If there is an action that can complete the task, perform it immediately. It's better if you can complete the task with fewer actions.
Your answer must be composed of the following five sections:
First, explain in detail what has been done so far up to action_1 and analyze why these steps were needed. Do not assume that the user could have made a mistake.

Secondly, describe every single screen component that contains information that helps user solve the task or that needs to be interacted with. Explain about the components step by step in a detailed manner considering the given task.
When the task deals with a list of items (e.g. finding the size of a group, identifying N-th item in order, etc.), you must include the full iteration of each and every items, like this: (1)first_item, (2)second_item, ..., (N)N-th_item.

Thirdly, describe what needs to be done to complete the task, detailing each action from start to finish. Then, identify which steps can currently be performed based on the UI elements visible on the screen, and mention the ID of these elements.
If there is a demo available, first describe the sequence of actions demonstrated, then link these actions to the steps in your plan before outlining the complete action plan.

Lastly, each action must be in the form of Action_2=(Action: functions.some_function_name, Argument: {property_name: property_val}).
Return the actions that need to be performed on the current screen.
The actions must be separated by new line characters.
In case there are three actions to be performed, your response will be in the following form: ´´´
Action_2=(Action: functions.some_function_name, Argument: {property_name: property_val})
Action_3=(Action: functions.some_function_name, Argument: {property_name: property_val})
Action_4=(Action: functions.some_function_name, Argument: {property_name: property_val})
´´´

- MAKE SURE that when you recommend actions that need to interact with UI elements, the UI elements MUST BE in the current screen.
- MAKE SURE that when satisfying the task completion conditions, click the 'Submit' button to finish.
- MAKE SURE that even if you do not find a suitable action, return the most plausible one.
- MAKE SURE that since texts in elements are extracted by recognizing with OCR, solve the given task considering that some mis-typos could exist.

Figure 6: An example of the CAAP prompt. The raionales contained in the expert demonstrations and the CoT-inducing instructions, which are detailed in the ' Instructions ' section, help an LLM gradually develop the semantic connectivity between the task description, the action history, and the visual state of the screen, in a CoT manner.

### Analysis of Action History
So far, the only action taken has been the initiation of the task (action_1: {name: start}). This initial action is necessary to begin the process of selecting "Helli" from the dropdown list and submitting the choice. No specific interactions with the UI elements have been made yet.

### Description of Screen Components
1. **Dropdown Menu (element_1)**: This element is a dropdown menu currently displaying "Theodora" as the selected item. Its coordinates are given as X: 76 [1-152], Y: 66 [55-76], indicating its position on the screen. The dropdown is not focused, meaning it hasn't been interacted with yet. This element is crucial for the task as it contains the list of names from which "Helli" needs to be selected.

2. **Submit Button (element_2)**: This is a button with the text "Submit" located at X: 50 [1-98], Y: 96 [80-112]. This button is essential for completing the task as it needs to be clicked after selecting "Helli" from the dropdown menu to confirm the selection.

### Completion Plan
To complete the task, the following steps need to be taken:
1. Click on the dropdown menu to expand it and reveal the list of names.
2. Select "Helli" from the list of names in the dropdown.
3. Click the "Submit" button to confirm the selection.

Based on the current screen, the actions that can be performed are:
- Clicking on the dropdown menu (element_1) to expand it and reveal the list of names, including "Helli".
- After selecting "Helli", clicking on the "Submit" button (element_2) will be necessary, but this can only be done after "Helli" has been successfully selected.

Given the current screen elements, the next action is to interact with the dropdown menu to expand it and reveal the list of names.

### Actions to be Performed
```
Action_2=(Action: functions.click_element, Argument: {element_id: 1})
```

Figure 7: An example of an LLM response to the CAAP prompt in Figure 8, showing how the LLM improves evidence for action-proposal by connecting the surrounding context information.

# E FUNCTION CALLING EXAMPLE

```
[
  {
    "name": "click_element",
    "description": "This action moves the mouse pointer to a
    screen element and performs a left-click to activate that
    element.",
    "parameters": {
      "type": "object",
      "properties": {
        "element_id": {
          "type": "integer",
          "description": "The id number (ranged from 1 to N) of
    the screen element to act on. A list of elements will be
    provided with the corresponding id numbers."
        }
      },
      "required": ["element_id"]
    }
  },
  {
    "name": "type_text",
    "description": "This action makes keyboard typing actions to
    enter the text, for example, into the 'input_field'-type
    screen element. Before typing the text, the element on which
    enter the text must be focused.",
    "parameters": {
      "type": "object",
      "properties": {
        "string_to_type": {
          "type": "string",
          "description": "The text to type"
        }
      },
      "required": ["string_to_type"]
    }
  }
]
```

Figure 8: The example provided illustrates the conversion of action type candidates into the input format supported by OpenAI API for function calling. As demonstrated in this example, information about action types can be conveyed through either of a function calling interface of an LLM service API or a prompt that includes the description of the candidates in plain text.

## F  TASK COVERAGE OF THE AGENTS THAT RELY SOLELY ON SCREEN IMAGES FOR MINIWOB++

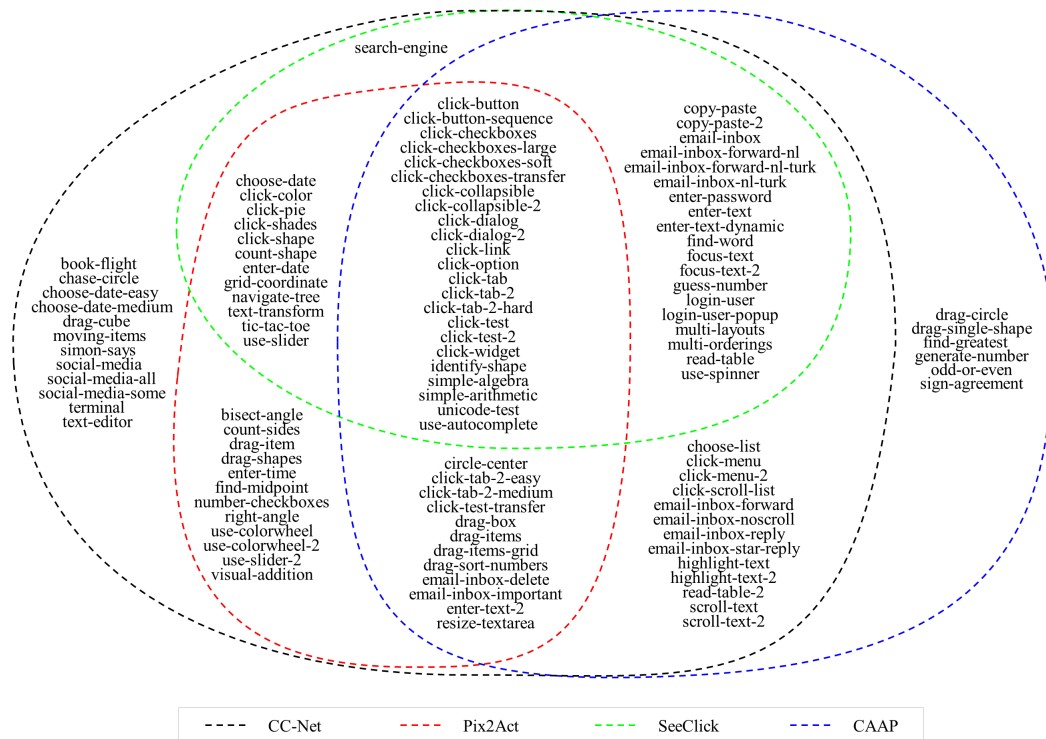

Figure 9: Task coverage of the agents with image-only inputs for MiniWoB++

## G  COMPARISON OF THE CAAP AGENT WITH SOTAS IN MINIWOB++ PER-TASK RESULT

Table 7: Comparison with other works in terms of SR per task for MiniWoB++ problems.

| Task | CAAP | SeeClick | Pix2Act | CC-Net | WebGUM | AdaPlanner | RCI | Human |
|---|---|---|---|---|---|---|---|---|
| bisect-angle | - | - | 0.960 | 0.970 | - | - | - | 0.920 |
| book-flight | - | - | - | 0.870 | 0.980 | - | - | 0.870 |
| chase-circle | - | - | - | 0.930 | - | - | - | 0.820 |
| choose-date | - | 0.020 | 0.790 | 0.970 | 1.000 | - | - | 0.970 |
| choose-date-easy | - | - | - | 0.990 | 1.000 | - | - | 0.990 |
| choose-date-medium | - | - | - | 0.990 | 1.000 | - | - | 0.980 |
| choose-list | 1.000 | - | - | 0.990 | 1.000 | 1.000 | 1.000 | 0.980 |
| circle-center | 1.000 | - | 0.960 | 0.970 | - | - | - | 0.960 |
| click-button | 0.980 | 0.960 | 0.990 | 1.000 | 1.000 | 1.000 | 1.000 | 0.980 |
| click-button-sequence | 0.920 | 0.860 | 0.990 | 1.000 | 1.000 | 1.000 | 1.000 | 0.940 |
| click-checkboxes | 0.980 | 0.780 | 1.000 | 0.980 | 1.000 | 1.000 | 1.000 | 0.970 |
| click-checkboxes-large | 1.000 | 0.020 | 0.990 | 0.710 | 0.990 | 1.000 | 0.940 | 0.870 |
| click-checkboxes-soft | 1.000 | 0.220 | 0.610 | 0.950 | 1.000 | 0.800 | 0.960 | 0.730 |
| click-checkboxes-transfer | 1.000 | 0.700 | 1.000 | 0.990 | 1.000 | 0.980 | 1.000 | 0.980 |
| click-collapsible | 0.920 | 1.000 | 0.940 | 1.000 | 1.000 | 1.000 | 1.000 | 0.990 |
| click-collapsible-2 | 0.900 | 0.480 | 0.970 | 0.980 | 0.950 | 0.840 | 1.000 | 0.970 |
| click-color | - | 1.000 | 0.990 | 1.000 | 1.000 | 1.000 | 1.000 | 0.970 |
| click-dialog | 1.000 | 1.000 | 1.000 | 1.000 | 1.000 | 1.000 | 1.000 | 1.000 |
| click-dialog-2 | 1.000 | 1.000 | 1.000 | 1.000 | 1.000 | 1.000 | 1.000 | 0.990 |
| click-link | 1.000 | 0.900 | 0.980 | 0.990 | 1.000 | 0.980 | 1.000 | 0.990 |
| click-menu | 0.800 | - | - | 0.940 | 0.970 | 0.780 | 1.000 | 0.970 |
| click-menu-2 | 0.920 | - | - | 0.830 | - | - | - | 0.980 |
| click-option | 0.960 | 1.000 | 1.000 | 0.990 | 1.000 | 1.000 | 1.000 | 0.990 |
| click-pie | - | 0.800 | 0.990 | 0.970 | 0.990 | - | - | 0.980 |
| click-scroll-list | 0.940 | - | - | 0.600 | 1.000 | 1.000 | 1.000 | 0.910 |
| click-shades | - | 0.020 | 0.990 | 1.000 | 1.000 | 1.000 | 1.000 | 0.910 |
| click-shape | - | 0.520 | 0.940 | 0.950 | 0.940 | 0.750 | 0.980 | 0.880 |
| click-tab | 1.000 | 1.000 | 1.000 | 1.000 | 1.000 | 1.000 | 1.000 | 0.990 |
| click-tab-2 | 0.960 | 0.600 | 0.980 | 0.980 | 0.990 | 0.850 | 1.000 | 0.970 |
| click-tab-2-easy | 1.000 | - | 0.990 | 0.990 | - | - | - | 0.990 |
| click-tab-2-hard | 0.980 | 0.420 | 0.970 | 0.980 | 0.950 | 0.780 | 0.980 | 0.960 |
| click-tab-2-medium | 1.000 | - | 1.000 | 0.990 | - | - | - | 0.970 |
| click-test | 1.000 | 1.000 | 1.000 | 1.000 | 1.000 | 1.000 | 1.000 | 1.000 |
| click-test-2 | 0.920 | 0.940 | 1.000 | 1.000 | 1.000 | 1.000 | 1.000 | 0.990 |
| click-test-transfer | 0.920 | - | 1.000 | 1.000 | - | - | - | 0.990 |
| click-widget | 1.000 | 0.580 | 1.000 | 1.000 | 1.000 | 1.000 | 0.980 | 0.830 |
| copy-paste | 0.980 | 0.800 | - | 0.790 | - | - | - | 0.940 |
| copy-paste-2 | 1.000 | 0.800 | - | 0.630 | - | - | - | 0.940 |
| count-shape | - | 0.280 | 0.700 | 0.850 | 0.680 | 0.500 | 0.400 | 0.820 |
| count-sides | - | - | 1.000 | 1.000 | - | - | - | 0.980 |
| drag-box | 1.000 | - | 0.990 | 1.000 | - | - | - | 0.990 |
| drag-circle | 0.980 | - | - | - | - | - | - | - |

| Task | CAAP | SeeClick | Pix2Act | CC-Net | WebGUM | AdaPlanner | RCI | Human |
|------|------|----------|---------|--------|--------|------------|-----|-------|
| drag-cube | - | - | - | 0.790 | - | - | - | 0.990 |
| drag-item | - | - | 1.000 | 1.000 | - | - | - | 0.980 |
| drag-items | 0.880 | - | 1.000 | 0.990 | - | - | - | 0.930 |
| drag-items-grid | 0.780 | - | 0.890 | 0.980 | - | - | - | 0.870 |
| drag-shapes | - | - | 0.980 | 0.990 | - | - | - | 0.960 |
| drag-single-shape | 0.940 | - | - | - | - | - | - | - |
| drag-sort-numbers | 0.720 | - | 0.950 | 0.970 | - | - | - | 0.920 |
| email-inbox | 0.960 | 0.800 | - | 1.000 | 1.000 | 0.980 | 0.980 | 0.960 |
| email-inbox-delete | 1.000 | - | 1.000 | 1.000 | - | - | - | 0.990 |
| email-inbox-forward | 0.960 | - | - | 1.000 | - | - | - | 0.960 |
| email-inbox-forward-nl | 0.960 | 0.740 | - | 1.000 | 1.000 | 1.000 | 1.000 | 0.910 |
| email-inbox-forward-nl-turk | 0.820 | 0.560 | - | 1.000 | 1.000 | 1.000 | 0.940 | 0.880 |
| email-inbox-important | 1.000 | - | 1.000 | 1.000 | - | - | - | 0.990 |
| email-inbox-nl-turk | 0.900 | 0.680 | - | 1.000 | 1.000 | 0.900 | 0.980 | 0.930 |
| email-inbox-noscroll | 0.980 | - | - | 1.000 | 0.000 | - | - | 0.960 |
| email-inbox-reply | 0.960 | - | - | 1.000 | - | - | - | 0.910 |
| email-inbox-star-reply | 0.980 | - | - | 1.000 | - | - | - | 0.950 |
| enter-date | - | 1.000 | 1.000 | 1.000 | 1.000 | 1.000 | 0.960 | 0.970 |
| enter-password | 1.000 | 1.000 | - | 1.000 | 1.000 | 0.980 | 1.000 | 0.960 |
| enter-text | 1.000 | 1.000 | - | 1.000 | 1.000 | 0.980 | 1.000 | 0.980 |
| enter-text-2 | 1.000 | - | 0.970 | 0.980 | - | - | - | 0.910 |
| enter-text-dynamic | 1.000 | 1.000 | - | 1.000 | 1.000 | 0.960 | 1.000 | 0.970 |
| enter-time | - | - | 1.000 | 0.970 | 1.000 | 0.960 | 1.000 | 0.980 |
| find-greatest | 1.000 | - | - | - | - | - | - | - |
| find-midpoint | - | - | 0.960 | 0.970 | - | - | - | 0.940 |
| find-word | 0.820 | 0.100 | - | 0.880 | - | - | - | 0.960 |
| focus-text | 1.000 | 1.000 | - | 1.000 | 1.000 | 1.000 | 1.000 | 1.000 |
| focus-text-2 | 1.000 | 0.960 | - | 1.000 | 1.000 | 0.940 | 1.000 | 0.990 |
| generate-number | 0.860 | - | - | - | - | - | - | - |
| grid-coordinate | - | 0.520 | 0.920 | 1.000 | 1.000 | 1.000 | 1.000 | 0.870 |
| guess-number | 0.980 | 1.000 | - | 1.000 | 0.430 | 0.880 | 0.200 | 0.990 |
| highlight-text | 0.920 | - | - | 1.000 | - | - | - | 0.970 |
| highlight-text-2 | 0.460 | - | - | 1.000 | - | - | - | 0.970 |
| identify-shape | 0.980 | 0.680 | 1.000 | 1.000 | 1.000 | 0.960 | 1.000 | 0.980 |
| login-user | 1.000 | 1.000 | - | 1.000 | 1.000 | 1.000 | 1.000 | 0.960 |
| login-user-popup | 1.000 | 0.980 | - | 1.000 | 1.000 | 0.980 | 0.680 | 0.940 |
| moving-items | - | - | - | 0.880 | - | - | - | 0.180 |
| multi-layouts | 0.980 | 0.720 | - | 1.000 | 1.000 | 0.840 | 0.960 | 0.950 |
| multi-orderings | 1.000 | 0.860 | - | 1.000 | 1.000 | 1.000 | 1.000 | 0.960 |
| navigate-tree | - | 0.820 | 0.990 | 0.990 | 1.000 | 0.820 | 1.000 | 0.980 |
| number-checkboxes | - | - | 0.840 | 0.990 | - | - | - | 0.960 |
| odd-or-even | 0.980 | - | - | - | - | - | - | - |
| read-table | 0.940 | 0.720 | - | 0.970 | - | - | - | 0.970 |
| read-table-2 | 0.740 | - | - | 0.940 | - | - | - | 0.950 |
| resize-textarea | 1.000 | - | 0.990 | 1.000 | - | - | - | 0.940 |
| right-angle | - | - | 0.970 | 0.980 | - | - | - | 0.870 |

| Task | CAAP | SeeClick | Pix2Act | CC-Net | WebGUM | AdaPlanner | RCI | Human |
|---|---|---|---|---|---|---|---|---|
| scroll-text | 0.620 | - | - | 0.960 | - | - | - | 0.970 |
| scroll-text-2 | 1.000 | - | - | 1.000 | - | - | - | 0.970 |
| search-engine | - | 0.840 | - | 1.000 | 0.960 | 1.000 | 1.000 | 0.970 |
| simon-says | - | - | - | 0.000 | - | - | - | 0.620 |
| simple-algebra | 1.000 | 0.380 | 1.000 | 0.750 | - | 0.820 | 1.000 | 0.860 |
| simple-arithmetic | 1.000 | 0.780 | 1.000 | 0.860 | - | - | - | 0.960 |
| sign-agreement | 0.960 | - | - | - | - | - | - | - |
| social-media | - | - | - | 0.900 | 1.000 | 0.820 | 0.980 | 0.960 |
| social-media-all | - | - | - | 0.750 | 0.520 | 1.000 | 1.000 | 0.890 |
| social-media-some | - | - | - | 0.850 | 0.730 | 0.900 | 0.960 | 0.910 |
| terminal | - | - | - | 0.000 | - | 0.980 | 1.000 | 0.880 |
| text-editor | - | - | - | 0.980 | - | - | - | 0.880 |
| text-transform | - | 0.460 | 0.920 | 0.600 | - | - | 0.800 | 0.860 |
| tic-tac-toe | - | 0.580 | 0.830 | 0.830 | 0.560 | 0.480 | 0.560 | 0.710 |
| unicode-test | 0.900 | 0.000 | 1.000 | 1.000 | - | - | - | 0.990 |
| use-autocomplete | 0.960 | 0.820 | 0.990 | 1.000 | 0.980 | 0.880 | 0.580 | 0.980 |
| use-colorwheel | - | - | 0.970 | 0.980 | - | - | - | 0.900 |
| use-colorwheel-2 | - | - | 0.950 | 0.950 | - | - | - | 0.940 |
| use-slider | - | 0.320 | 0.920 | 0.910 | - | - | - | 0.980 |
| use-slider-2 | - | - | 1.000 | 0.950 | - | - | - | 0.970 |
| use-spinner | 0.980 | 0.160 | - | 1.000 | 0.110 | 0.900 | 0.960 | 0.980 |
| visual-addition | - | - | 1.000 | 0.990 | - | - | - | 0.970 |
| Average SR | 0.945 | 0.694 | 0.962 | 0.936 | 0.925 | 0.929 | 0.940 | 0.935 |

# H ANNOTATION TOOL

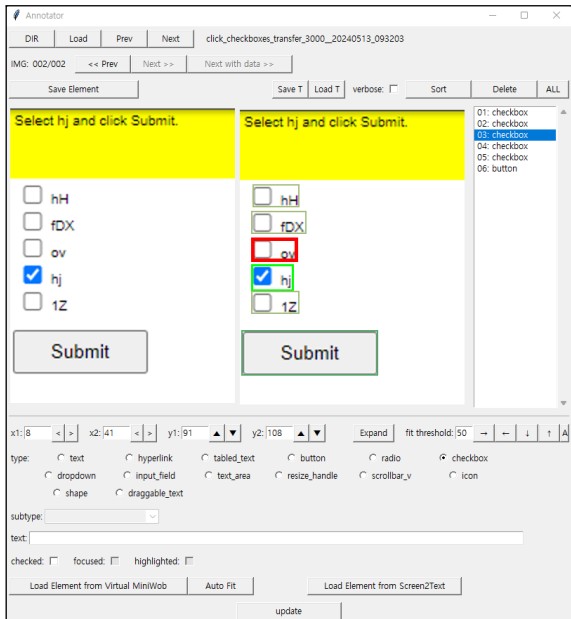

Figure 10: Our GUI-based annotation tool enables efficient annotation of UI elements within screenshots.

# I COMPARISONS TO OTHER APPROACHES

Table 8 summarizes the input types, agent architectures, training methodologies, and whether experimental results on MiniWoB++ and WebShop are reported for our method and other approaches.

Table 8: Comparison of agent configurations in input types, architectures, training methodologies.

| Method | Modality | Model | | Training | Experiment | |
| | | Vision | Decision | Scheme | MiniWoB++ | WebShop |
|---|---|---|---|---|---|---|
| Human | - | - | - | - | - | |
| WGE | DOM | - | DOM-NET | SL+RL | √ | - |
| WebN-T5 | HTML | - | WebN-T5-3B | SL | √ | - |
| RCI | HTML | - | gpt-4 | - | √ | - |
| AdaPlanner | HTML | - | text-davinci-003 | - | √ | - |
| WebShop | HTML+Image | ResNet | BERT, BART | SL+RL | - | √ |
| WebGUM | HTML+Image | ViT-B16 | Flan-T5-XL | SL | √ | √ |
| CC-Net | DOM+Image / Image only | ResNet | Transformer | SL+RL | √ | - |
| Pix2Act | Image | Pix2Struct | Transformer | SL+RL | √ | √ |
| SeeClick | Image | ViT | Qwen-VL 9.6B | SL | √ | - |
| CAAP | Image | YOLOv8, Pix2Struct | gpt-4-0125 | SL | √ | √ |

## I.1 EXPERIMENT RESULT FOR MINIWOB++

The data requirements and reported performance in literature are described in Table 9. One notable approach is CC-Net, which reports results across the broadest range of MiniWoB++ tasks. However, this required the collection of 2.4 million demonstrations, demanding approximately 6,300 hours of human labor, and faces challenges in real-world applications due to its high reliance on DOM information. RCI (Kim et al., 2023) and AdaPlanner (Sun et al., 2023) achieved average task SRs above 92% with fewer than 100 demonstrations, but their dependence on HTML inputs limits their task coverage. When recalculating SRs, including unsupported or unreported tasks, their performance drops to 50.8% and 49.2%, respectively. WebGUM (Furuta et al., 2023) expanded task coverage by using both HTML and image inputs. However, despite utilizing 401K demonstrations, its results only showed a 1.9% improvement over RCI, achieving an SR of 52.7%. Pix2Act (Shaw et al., 2023) was the first method to achieve high task SRs on MiniWoB++ using image inputs exclusively. It leveraged 1.3 million demonstrations, achieving over 96% success on 58 tasks, with an SR of 55.8% across the same set of 100 tasks. In contrast, SeeClick, which also used image inputs but on a smaller demonstration dataset, failed to sufficiently demonstrate problem-solving ability. Our approach represents the most advanced image-only agent, solving 73 tasks with a high SR of 94.5%, using fewer than 1.8K demonstrations. Furthermore, it achieved a 63.3% SR on the 100-task range, 7.5% higher than Pix2Act, with 61 tasks demonstrating an SR of over 80%—five more tasks than Pix2Act. (See Table 7 of Appendix G.) This represents a significant improvement achieved with fewer than 100 human demonstrations, resulting from the powerful reasoning capabilities of the LLM and our carefully modularized agent design.

## I.2 EXPERIMENT RESULT FOR WEBSHOP

Table 10 compares our approach with studies reporting performance on the WebShop benchmark. WebGUM was trained using both HTML and image inputs with 1K human demonstrations, achieving an average

Table 9: Comparison with other methods that solve MiniWoB++ problems. While the primary configuration of CC-Net utilizes the DOM+Image modality, we have also included the performance of the image-only modality in this table to provide a more in-depth comparison.

| Method | Modality | Image datasize | Demo datasize | Reported SR | Reported tasks |
|---|---|---|---|---|---|
| Human | - | - | - | 93.5% | 104 |
| WGE | DOM | 0 | ~4.8K | 66.5% | 48 |
| WebN-T5 | HTML | 0 | 12K | 48.4% | 56 |
| RCI | HTML | 0 | ~0.1K | 94.0% | 54 |
| AdaPlanner | HTML | 0 | ~0.1K | 92.9% | 53 |
| WebGUM | HTML+Image | 0 | 401K | 94.2% | 56 |
| CC-Net | DOM+Image | 0 | 2.4M | 93.6% | 104 |
| CC-Net | Image | 0 | 2.4M | 24.0% | 104 |
| Pix2Act | Image | 0 | 1.3M | 96.2% | 59 |
| SeeClick | Image | 0 | 2.8K | 69.4% | 55 |
| CAAP | Image | 1.76K | 0.1K | 94.5% | 73 |

reward of 66.6, surpassing the performance of the initial model introduced with the WebShop benchmark, which was trained using an Supervised Learning (SL)+Reinforcement Learning (RL) approach. Pix2Act, designed to operate solely on image inputs, reported an average reward of 46.7 when trained with 1K human demonstrations using a combination of SL and RL techniques. Our agent, utilizing a vision model training dataset collected through an automated process and just a single human demonstration, achieved a significantly higher average reward of 62.3, outperforming Pix2Act. This result is comparable to methods that leverage both HTML and image inputs, demonstrating that our approach can effectively function across web and non-web environments with minimal human effort in dataset collection.

Table 10: Comparison with other methods that solve WebShop problems.

| Method | Modality | Image datasize | Demo datasize | Task score | SR |
|---|---|---|---|---|---|
| Human | - | - | - | 82.1 | 59.6% |
| WebShop | HTML+Image | 0 | 10.6K | 62.4 | 28.7% |
| WebGUM | HTML+Image | 0 | 1K | 67.5 | 45.0% |
| Pix2Act | Image | 0 | 1K | 46.7 | N/A |
| CAAP | Image | 0.3K | 1 | 62.3 | 34.7% |

## J  PER-TASK RESULTS FOR ABLATION STUDY ON CAAP COMPONENTS

Table 11: Per-task SRs with respect to varying combinations of prompt components.

| Task | Full configurations | Demo without CoT phrases | CoT phrases without demo | CoT phrases with action-only demo | No CoT, no demo |
|---|---|---|---|---|---|
| choose-list | 1.000 | 0.980 | 0.940 | 0.920 | 0.980 |
| circle-center | 1.000 | 1.000 | 1.000 | 0.980 | 0.980 |
| click-button | 0.980 | 0.980 | 0.980 | 0.980 | 0.980 |
| click-button-sequence | 0.920 | 0.920 | 0.920 | 0.920 | 0.920 |
| click-checkboxes | 0.980 | 1.000 | 1.000 | 1.000 | 0.960 |
| click-checkboxes-large | 1.000 | 0.960 | 1.000 | 1.000 | 0.860 |
| click-checkboxes-soft | 1.000 | 0.980 | 0.980 | 1.000 | 0.960 |
| click-checkboxes-transfer | 1.000 | 1.000 | 1.000 | 1.000 | 1.000 |
| click-collapsible | 0.920 | 0.140 | 1.000 | 0.960 | 0.340 |
| click-collapsible-2 | 0.900 | 0.900 | 0.920 | 0.920 | 0.780 |
| click-dialog | 1.000 | 1.000 | 1.000 | 1.000 | 1.000 |
| click-dialog-2 | 1.000 | 1.000 | 0.980 | 1.000 | 1.000 |
| click-link | 1.000 | 1.000 | 1.000 | 1.000 | 1.000 |
| click-menu | 0.800 | 0.880 | 0.360 | 0.400 | 0.360 |
| click-menu-2 | 0.920 | 1.000 | 0.560 | 0.660 | 0.780 |
| click-option | 0.960 | 0.960 | 0.960 | 0.980 | 0.940 |
| click-scroll-list | 0.940 | 0.940 | 0.860 | 0.920 | 0.700 |
| click-tab | 1.000 | 1.000 | 1.000 | 1.000 | 1.000 |
| click-tab-2 | 0.960 | 0.980 | 0.980 | 0.960 | 0.980 |
| click-tab-2-easy | 1.000 | 1.000 | 1.000 | 1.000 | 1.000 |
| click-tab-2-hard | 0.980 | 1.000 | 1.000 | 1.000 | 1.000 |
| click-tab-2-medium | 1.000 | 1.000 | 1.000 | 1.000 | 1.000 |
| click-test | 1.000 | 1.000 | 1.000 | 1.000 | 1.000 |
| click-test-2 | 0.920 | 0.920 | 0.920 | 0.920 | 0.920 |
| click-test-transfer | 0.920 | 0.920 | 0.920 | 0.920 | 0.920 |
| click-widget | 1.000 | 0.960 | 0.860 | 0.860 | 0.860 |
| copy-paste | 0.980 | 0.980 | 0.980 | 0.980 | 0.960 |
| copy-paste-2 | 1.000 | 1.000 | 0.820 | 0.980 | 0.780 |
| drag-box | 1.000 | 1.000 | 0.980 | 1.000 | 0.960 |
| drag-circle | 0.980 | 0.980 | 0.980 | 0.980 | 0.780 |
| drag-items | 0.880 | 0.860 | 0.940 | 0.940 | 0.580 |
| drag-items-grid | 0.780 | 0.620 | 0.600 | 0.780 | 0.620 |
| drag-single-shape | 0.940 | 0.940 | 0.940 | 0.900 | 0.740 |
| drag-sort-numbers | 0.720 | 0.480 | 0.560 | 0.620 | 0.160 |
| email-inbox | 0.960 | 0.960 | 0.860 | 0.900 | 0.720 |
| email-inbox-delete | 1.000 | 1.000 | 1.000 | 1.000 | 0.980 |
| email-inbox-forward | 0.960 | 0.980 | 0.640 | 0.960 | 0.620 |
| email-inbox-forward-nl | 0.960 | 1.000 | 0.460 | 0.920 | 0.520 |
| email-inbox-forward-nl-turk | 0.820 | 0.800 | 0.220 | 0.820 | 0.380 |
| email-inbox-important | 1.000 | 1.000 | 0.980 | 1.000 | 0.960 |
| email-inbox-nl-turk | 0.900 | 0.800 | 0.540 | 0.860 | 0.500 |

| Task | Full configurations | Demo without CoT phrases | CoT phrases without demo | CoT phrases with action-only demo | No CoT, no demo |
|---|---|---|---|---|---|
| email-inbox-noscroll | 0.980 | 0.940 | 0.920 | 1.000 | 0.860 |
| email-inbox-reply | 0.960 | 0.980 | 0.940 | 0.900 | 0.960 |
| email-inbox-star-reply | 0.980 | 0.960 | 0.980 | 0.960 | 0.960 |
| enter-password | 1.000 | 1.000 | 1.000 | 1.000 | 0.980 |
| enter-text | 1.000 | 1.000 | 1.000 | 1.000 | 0.980 |
| enter-text-2 | 1.000 | 1.000 | 1.000 | 1.000 | 0.980 |
| enter-text-dynamic | 1.000 | 1.000 | 1.000 | 1.000 | 0.980 |
| find-greatest | 1.000 | 1.000 | 0.140 | 0.900 | 0.140 |
| find-word | 0.820 | 0.620 | 0.740 | 0.800 | 0.380 |
| focus-text | 1.000 | 1.000 | 1.000 | 1.000 | 0.980 |
| focus-text-2 | 1.000 | 1.000 | 1.000 | 1.000 | 0.980 |
| generate-number | 0.860 | 0.800 | 0.840 | 0.760 | 0.660 |
| guess-number | 0.980 | 1.000 | 0.860 | 0.960 | 0.900 |
| highlight-text | 0.920 | 0.960 | 0.680 | 0.900 | 0.940 |
| highlight-text-2 | 0.460 | 0.500 | 0.140 | 0.400 | 0.240 |
| identify-shape | 0.980 | 0.980 | 0.940 | 0.960 | 0.960 |
| login-user | 1.000 | 1.000 | 1.000 | 1.000 | 0.980 |
| login-user-popup | 1.000 | 0.820 | 0.440 | 0.440 | 0.440 |
| multi-layouts | 0.980 | 0.920 | 0.960 | 1.000 | 0.660 |
| multi-orderings | 1.000 | 1.000 | 0.960 | 1.000 | 0.660 |
| odd-or-even | 0.980 | 0.860 | 0.980 | 0.960 | 0.840 |
| read-table | 0.940 | 0.820 | 0.900 | 0.900 | 0.840 |
| read-table-2 | 0.740 | 0.700 | 0.720 | 0.720 | 0.700 |
| resize-textarea | 1.000 | 1.000 | 0.900 | 0.980 | 0.980 |
| scroll-text | 0.620 | 0.780 | 0.200 | 0.420 | 0.020 |
| scroll-text-2 | 1.000 | 0.980 | 0.720 | 0.880 | 0.780 |
| simple-algebra | 1.000 | 0.980 | 1.000 | 1.000 | 0.700 |
| simple-arithmetic | 1.000 | 1.000 | 1.000 | 1.000 | 0.940 |
| sign-agreement | 0.960 | 0.920 | 0.420 | 0.920 | 0.420 |
| unicode-test | 0.900 | 0.920 | 0.900 | 0.880 | 0.880 |
| use-autocomplete | 0.960 | 0.880 | 0.900 | 0.960 | 0.900 |
| use-spinner | 0.980 | 1.000 | 0.900 | 0.920 | 0.700 |
| Average SR | 0.945 | 0.920 | 0.845 | 0.910 | 0.792 |

# K   COMPARISON OF THE CAAP AND RCI PROMPTINGS

Table 12: Per-task SRs for RCI agent with CAAP and original RCI agent.

| Task | RCI agent with CAAP | RCI agent (original) | Task | RCI agent with CAAP | RCI agent (original) |
|---|---|---|---|---|---|
| choose-list | 0.80 | 1.00 | email-inbox | 0.66 | 0.82 |
| click-button | 1.00 | 1.00 | email-inbox-forward-nl | 0.90 | 0.02 |
| click-button-sequence | 1.00 | 1.00 | email-inbox-forward-nl-turk | 1.00 | 0.20 |
| click-checkboxes | 1.00 | 1.00 | email-inbox-nl-turk | 0.40 | 0.28 |
| click-checkboxes-large | 1.00 | 1.00 | enter-date | 0.00 | 0.00 |
| click-checkboxes-soft | 0.98 | 0.84 | enter-password | 1.00 | 1.00 |
| click-checkboxes-transfer | 1.00 | 1.00 | enter-time | 0.02 | 0.00 |
| click-collapsible | 1.00 | 1.00 | focus-text | 0.98 | 1.00 |
| click-collapsible-2 | 0.92 | 0.82 | grid-coordinate | 0.96 | 1.00 |
| click-color | 1.00 | 1.00 | identify-shape | 1.00 | 0.76 |
| click-dialog | 1.00 | 1.00 | login-user-popup | 0.44 | 0.44 |
| click-dialog-2 | 1.00 | 1.00 | multi-layouts | 0.88 | 0.82 |
| click-link | 0.98 | 1.00 | navigate-tree | 0.92 | 0.86 |
| click-menu | 1.00 | 1.00 | search-engine | 0.30 | 0.30 |
| click-option | 1.00 | 1.00 | simple-algebra | 0.80 | 0.40 |
| click-scroll-list | 1.00 | 1.00 | social-media | 0.98 | 0.96 |
| click-shades | 1.00 | 1.00 | social-media-all | 0.98 | 0.98 |
| click-shape | 0.94 | 0.94 | social-media-some | 0.90 | 0.44 |
| click-tab | 1.00 | 1.00 | terminal | 0.00 | 0.00 |
| click-tab-2 | 0.92 | 0.94 | tic-tac-toe | 0.30 | 0.14 |
| click-tab-2-hard | 0.98 | 0.96 | use-autocomplete | 0.92 | 0.48 |
| click-test | 1.00 | 1.00 | use-spinner | 0.88 | 1.00 |
| click-test-2 | 0.98 | 1.00 | | | |
| click-widget | 1.00 | 1.00 | Average SR | 0.840 | 0.769 |
| count-shape | 0.78 | 0.76 | | | |

## L    CASE STUDY OF MINIWOB++ TASKS

In this section, we closely examine the failure cases that arise during the execution of MiniWoB++ tasks by the CAAP agent. Figure 11 presents screenshots of selected exemplary failure cases. We have categorized these eight instances into three groups: incorrect directives, observation failures, and action-proposal failures.

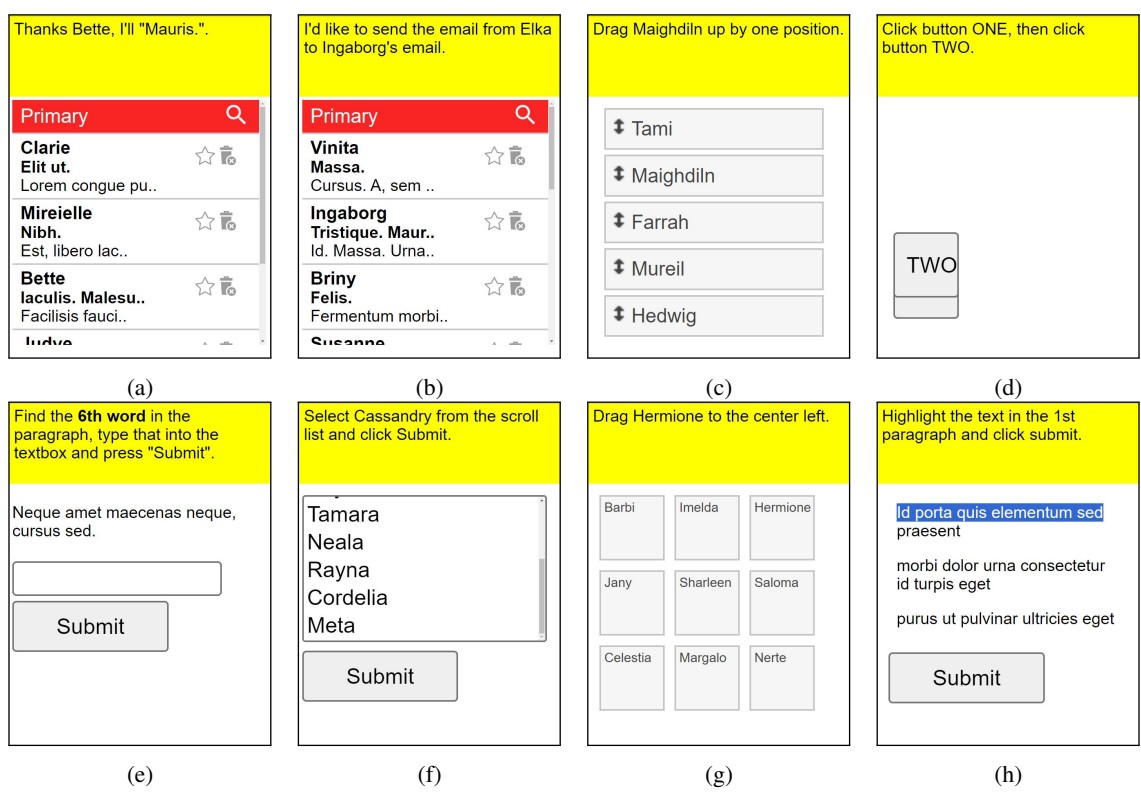

Figure 11: Fail cases of the CAAP agent.

**Incorrect directives**    In Figure 11, cases (a) and (b) correspond to failures due to incorrect directives. Case (a) involves a directive that is not a task command, exemplified by the nonsensical sentence "Thanks Bette, I'll Mauris." Thus, this case is a MiniWoB++ benchmark error. Case (b) concerns a directive with a grammatical error. In the directive "I'd like to send the email from Elka to Ingaborg's email," a human might recognize the grammatical mistake and deduce that the email should be forwarded to Ingaborg. However, in our failure case, the LLM inputs "Ingaborg's email." into the field designated for the forward recipient, instead, and attempts to send the email.

**Observation failures**    In Figure 11, cases (c) and (d) exemplify failures due to observation errors. Case (c) involves a failure where the task required identifying an object labeled 'Maighdiln'. However, during the UI element understanding process, the text was misrecognized as 'Maighdin' and conveyed it to the LLM, which incorrectly determined that the target object was absent, leading to failure. In case (d), the button marked 'ONE' is obscured behind a button labeled 'TWO'. Although the 'ONE' text is not visible in the screenshot,

the presence of the obscured button can be visually confirmed. A human operator would likely attempt to move the 'TWO' button to reveal and verify the text on the obscured button.

**Action-proposal failures**    In Figure 11, cases (e), (f), (g), and (h) illustrate failures resulting from action-proposal errors. Case (e) exemplifies a task requiring the identification of the sixth word, 'sed.' Although the visual observer correctly observed and reported the scene, the LLM incorrectly identified the sixth word. In case (f), the task was to find 'Cassandry', which was not visible on the screen. The LLM initiated a progressive scroll but prematurely stopped scrolling and failed to make the correct decision to continue. Case (g) involved the LLM determining the coordinates of text-written blocks and predicting their positions to move an object labeled 'Hermione' to the central left position. The LLM decided on the drag action but incorrectly positioned the endpoint, leading to failure. Case (h) required highlighting a specific paragraph within a document composed of multiple paragraphs. Our visual observer is designed to detect text on a line-by-line basis, making it a challenging task to predict paragraph structure from the coordinates of each line. In this case, our agent failed to accurately predict the paragraph configuration. These four cases represent failures due to incorrect judgments by the LLM, with particular difficulty, as seen in (g) and (h), in interpreting coordinates represented numerically.

Among the failure cases, a tiny portion of the cases result from errors in the benchmark directives. While roughly 40% arise due to observation errors inherent in the agent configuration based on visual observation data, the others are failures that occur within the action-proposal domain. However, most minor errors in directives or observations tend to be appropriately accounted for during the action-proposal process by the LLM, often leading to successful outcomes as much as failures. This tendency is expected to intensify as the reasoning capabilities of LLMs advance, thereby improving their task-solving abilities.

