# OpenReview forum: "CAAP: Context-Aware Action Planning Prompting to Solve Computer Tasks with Front-End UI Only"
_ICLR.cc/2025/Conference — Submitted to ICLR 2025_

### Official Review · Reviewer_99s6 · 2024-10-21

**Soundness:** 2
**Presentation:** 2
**Contribution:** 2
**Rating:** 5
**Confidence:** 3

**Summary:**

This paper introduces a modular agent to solve computer tasks using screenshot images (that are converted into text), Large Language Models,  and a Graphical User Interface. UI elements are first detected and formatted into text using fine-tuned models. Then, a large language model decides appropriate actions using a specific prompting technique. Finally, the actions are executed using the keyboard and mouse interfaces. As such, the model doesn't rely on HTML or the Document Object Model as input sources. The proposed prompting technique, CAAP, structures the available context and introduces chain-of-thought reasoning. Experiments on MiniWoB++ and WebShop benchmarks show good performance with a limited number of expert demonstrations.

**Strengths:**

- The paper addresses an interesting problem of solving computer tasks using LLMs and visual inputs.
- The engineering effort of integrating different modules into a unified framework is well-executed and appreciated.
- The provided implementation and experimental setup details are a valuable part of the paper.

**Weaknesses:**

- It is not clear why the “UI element understanding” module needs to understand the spatial relationships between the element and its adjacent elements (line 173). It seems that only the element's position is extracted. This should be clarified in the paper.
- The proposed prompting technique is described only superficially. Examples and more detailed discussions about each part of the prompt should be included in the main paper.
- The proposed prompting technique seems to be quite tailored towards a specific application and tested for a single language model. Therefore, it comes off as a hack rather than a novel contribution. Overall, the paper has limited scientific contribution. I would suggest the authors to focus more on investigating the influence the proposed technique has on In-Context Learning overall and highlighting their contribution in this way.
- The paper would benefit from a discussion about using VLMs, rather than a separate vision module and an LLM. Could a VLM be used to complete tasks in full or to replace the vision module, eliminating the need for fine-tuning different vision models?
- The comparison to different methods evaluated on different numbers of tasks doesn’t hold a lot of scientific value. Maybe the metrics only for the tasks all methods were evaluated on could be reported?
- The success rate range in Figure 4 (x-axis) overemphasises the performance of the proposed method when compared to ablation variants. Adjusting the range of choosing a different format for the graph could fix this issue.

**Questions:**

- What other applications could CAAP be applied to?
- On line 343, it is stated that some tasks were excluded from the agent’s training. How are other tasks used for training? Aren’t these demonstrations just used at inference as part of the prompt?
- What is the advantage of using a semi-masked screenshot rather than a fully masked or cropped screenshot of the element?

---

> ### Author Response · Authors · 2024-11-20
> **Response to the weaknesses: part 1**
>
> __Weakness 1) It is not clear why the “UI element understanding” module needs to understand the spatial relationships between the element and its adjacent elements (line 173). It seems that only the element's position is extracted. This should be clarified in the paper.__
>
> → The surrounding images of an UI element often contain valuable information that aids in accurately describing the target UI element. For instance, when dealing with text elements, focusing solely on the shape of the UI element without considering its surrounding context makes it difficult to distinguish whether the text is part of a button, an input field, or plain text within a table. Similarly, identifying whether blue text represents a simple label or a hyperlink can also pose challenges. However, when surrounding images are included, it becomes intuitively evident that they provide significant assistance in making precise judgments. We focused on enhancing the visual observer's capability to understand UI elements by employing a semi-masking approach, where the surrounding image is provided but darkened to emphasize the target element.
>
> Additionally, the position information, represented by X and Y coordinates, is extracted by the UI element detection submodule rather than the UI element understanding submodule of the visual observer. The information extracted by the UI element understanding submodule includes all output values depicted in Figure 2(c), except for the X and Y coordinates. To avoid any potential confusion, we will revise the caption of Figure 2 accordingly.
>
>
> __Weakness 2) The proposed prompting technique is described only superficially. Examples and more detailed discussions about each part of the prompt should be included in the main paper.__
>
> → In CAAP prompting, the emphasis lies not on the individual words themselves but on how the content is structured and delivered to the LLM in a specific sequence. Therefore, rather than focusing on the exact words in the prompt, we included Figure 3 in the main paper to illustrate the composition and flow of the content. Questions regarding more detailed discussions about CAAP prompting can be addressed by our officially provided comment, “Analysis of what traits help the CAAP prompting,” which we submitted in response to Reviewer tUvm’s feedback. We will ensure that this explanation is clearly reflected in the main paper.
>
>
> __Weakness 3) The proposed prompting technique seems to be quite tailored towards a specific application and tested for a single language model. Therefore, it comes off as a hack rather than a novel contribution. Overall, the paper has limited scientific contribution. I would suggest the authors to focus more on investigating the influence the proposed technique has on In-Context Learning overall and highlighting their contribution in this way.__
>
> → We would like to ask for clarification on the basis of the claim that CAAP prompting is "quite tailored" to specific applications. We strongly emphasize that no prompt customizations were made to improve performance on MiniWoB++ and WebShop tasks.
>
> Furthermore, there appears to be a significant misunderstanding in limiting our contribution solely to the CAAP prompting approach. Our primary contribution is the introduction of the first modular architecture-based GUI agent as an alternative for addressing challenges that are difficult to solve with VLM (LMM) alone. This is presented as the first key contribution of our work. Within this modular architecture, leveraging an LLM as an action planner required an effective prompting strategy to maximize inference performance, leading to the proposal of CAAP prompting. While both contributions should have been evenly emphasized throughout the paper, due to page limitations, we allocated significant focus to CAAP prompting in Sections 4 and 5.3.
>
> The reason for not conducting experiments with multiple language models is that our method is not tailored to any specific language model architecture. Thus, comparing performance across multiple language models would not provide meaningful insights to strengthen our claims and was deemed redundant. We believe that demonstrating the effectiveness of our approach with a single well-performing language model is sufficient to substantiate our contribution. Accordingly, we validated our method using the most up-to-date GPT model available at the time of our experiments. Instead, we prioritized comparative evaluations of the prompting strategies themselves. By comparing CAAP prompting with RCI prompting using the same language model, we demonstrated the superiority of our approach.

---

> > ### Author Response · Authors · 2024-11-20
> > **Response to the weaknesses: part 2**
> >
> > __Weakness 4) The paper would benefit from a discussion about using VLMs, rather than a separate vision module and an LLM. Could a VLM be used to complete tasks in full or to replace the vision module, eliminating the need for fine-tuning different vision models?__
> >
> > → At present, whether VLMs are used purely as vision models or as integrated modules for both screenshot understanding and action planning, fine-tuning remains indispensable. For a GUI agent to execute actions, such as clicks proposed by LLMs/VLMs, it must accurately estimate the pixel coordinates of the target elements on screen. This requirement is closely tied to the visual grounding problem, a critical topic in GUI agent research.
> >
> > Even the latest general-purpose VLMs, such as GPT-4V and Claude 3.5 Sonnet, demonstrate reasonable capabilities in image interpretation but frequently fail to accurately predict the precise pixel coordinates of objects within an image. As a result, regardless of how VLMs are employed, addressing this issue necessitates some degree of fine-tuning.
> >
> > Our approach tackles this challenge by leveraging well-established object detection techniques with proven performance, integrating them into a modular architecture. By implementing this solution, we have successfully demonstrated an alternative that can serve as a valuable reference for future research in GUI agents.
> >
> >
> > __Weakness 5) The comparison to different methods evaluated on different numbers of tasks doesn’t hold a lot of scientific value. Maybe the metrics only for the tasks all methods were evaluated on could be reported?__
> >
> > → Previous studies have often compared technical superiority based on success rates across different task scopes. However, we agree with you that this approach is inherently unfair. To address this, we conducted a more comprehensive comparative analysis that does not rely solely on reported success rates.
> >
> > As a potential alternative, we considered comparing success rates across the common set of tasks shared by all methods. However, since the number of shared tasks is limited to only 23, and the average accuracy improvement ranges between just 0.7% and 2.7%, we concluded that this approach also fails to provide meaningful scientific value.
> >
> > Consequently, instead of comparing methods based on their reported success rates, we determined that a more meaningful approach is to evaluate them comprehensively by examining structural differences, the diversity of task types they can handle, and the data collection costs required to extend them to new problems. Traces of our comprehensive comparative analysis can be found in Table 1 and Table 2 of the main paper, as well as in Table 4 and Table 5 of Appendix B.
> >
> >
> > __Weakness 6) The success rate range in Figure 4 (x-axis) overemphasises the performance of the proposed method when compared to ablation variants. Adjusting the range of choosing a different format for the graph could fix this issue.__
> >
> > → We utilized commonly used bar graphs to represent the data. However, due to the skewed distribution of the data, the readability was compromised. To address this, we zoomed in on the x-axis range where differences occur, specifically between 75% and 95%, to present the comparison results more clearly. This is a widely used visualization technique.

---

> ### Author Response · Authors · 2024-11-20
> **Response to the questions**
>
> __Q1) What other applications could CAAP be applied to?__
>
> → As indicated by the paper title, the CAAP agent is a GUI agent designed to handle general computer tasks. Specifically, the CAAP agent is designed to operate across a wide range of computer tasks that involve screenshots and mouse/keyboard interactions, such as web navigation, email replies, calendar scheduling, terminal operations, and OS environment setup. While the paper focuses on the MiniWoB++ and WebShop benchmarks, our primary goal was not to optimize performance on these specific benchmarks. Instead, we aimed to propose a general architecture for a GUI agent capable of addressing diverse computer tasks and provided solutions to key challenges to ensure the effective functioning of this architecture.
>
>
> __Q2) On line 343, it is stated that some tasks were excluded from the agent’s training. How are other tasks used for training? Aren’t these demonstrations just used at inference as part of the prompt?__
>
> → We deal with two types of data in our study. The first type consists of human demonstrations used as part of the prompt during inference, and the second type comprises training data for fine-tuning the visual observer. Since "agent's training," as mentioned in the paper, refers to parameter adjustment of the model, the content in line 343 pertains specifically to the fine-tuning of the visual observer.
>
> For collecting fine-tuning data for the visual observer, we followed a specific procedure for all tasks except six—drag-circle, drag-single-shape, find-greatest, generate-number, odd-or-even, and sign-agreement. First, we distinguished between seed sets for training data collection and those for testing. Next, for each task, simulations were sequentially performed using 10 seeds within the training seed range. Subsequently, we captured screenshots and performed annotations using a semi-automated process with a custom annotation tool described in Appendix G, Figure 6. The resulting dataset was then used to fine-tune the visual observer.
>
>
> __Q3) What is the advantage of using a semi-masked screenshot rather than a fully masked or cropped screenshot of the element?__
>
> → The semi-masking approach includes surrounding image information that cannot be provided by fully masking methods or by cropping only the target UI element. For a detailed explanation of the role that surrounding image information plays, please refer to our response to weakness 1.

---

> > ### Comment · Reviewer_99s6 · 2024-11-22
> > **Response to the rebuttal.**
> >
> > Thank you for your response and clarifications. Based on your response about the contributions, I have updated my original assessment regarding the contribution.
> >
> > Regarding validating the proposed prompting technique. You say "Thus, comparing performance across multiple language models would not provide meaningful insights to strengthen our claims and was deemed redundant." I disagree with this statement. Just because a prompting technique works for one model, doesn't mean it works for all (or most) of the models. It could be that it is only beneficial for this one type of LLM, without evaluating different models it is impossible to say.
> >
> > Regarding using a common set of tasks shared by all methods for evaluation. You say "...the average accuracy improvement ranges between just 0.7% and 2.7%, we concluded that this approach also fails to provide meaningful scientific value." Just because an experiment doesn't show a significant improvement for your method doesn't mean the results should be dismissed.
> >
> > Unless I missed it, the paper doesn't seem to be updated with the changes you said you'd incorporate in the rebuttal.

---

> ### Author Response · Authors · 2024-11-29
> **Response to the response**
>
> We hope our previous response has addressed your questions and concerns. We would now like to respond to your remaining concerns:
>
> **1) Verification of the CAAP prompting with multiple LLMs**
> We are currently running the WebShop experiments using Anthropic Claude Sonnet 3.5 and Llama-3.1-70B. Below, we provide the results for Claude Sonnet 3.5, and we will update you with the Llama results once they are complete.
>
> | Method                   | Image Data Size | Demo Data Size | Task Score |
> |--------------------------|-----------------|----------------|------------|
> | **CAAP (Claude Sonnet 3.5)** | 0.3K       | 1              | **68.0**      |
> | **CAAP (GPT-4)**        | 0.3K            | 1              | 62.3       |
> | **Pix2Act**             | 0               | 1K             | 46.7       |
> | **Human**               | -               | -              | 82.1       |
>
> As seen in the table, Sonnet 3.5 achieves improved results compared to GPT-4, demonstrating that our CAAP prompting method works effectively across different types of LLMs.
>
>
> **2) Results for the 23 Common MiniWob++ Tasks**
>
> We apologize for the ambiguity in our previous response. When we mentioned that the average accuracy improvement was minor, we were referring to the improvements in tested accuracy compared to the values obtained using the larger set of tasks already reported in our paper. Specifically, we meant that the difference is not significant. The table below provides a detailed comparison to clarify our point:
>
> | Method               | $SR_{reported}$ | $SR_{common}$ | $SR_{reported} - SR_{common}$ |
> |----------------------|-------------------------|-----------------------|-------------------------------|
> | **CAAP**             | 94.5 (73)               | 97.2 (23)             | +2.7                          |
> | **SeeClick**         | 69.4 (55)               | 70.1 (23)             | +0.7                          |
> | **Pix2Act**          | 96.2 (59)               | 97.4 (23)             | +1.2                          |
> | **CC-Net (no DOM)**  | 24.0 (104)              | N/A                   | N/A                           |
> | **Human**            | 93.5 (104)              | 95.3 (23)             | +1.8                          |
>
> - The numbers in parentheses indicate the number of tasks solved.
>
>
>
> Because the common set consists of only 23 tasks (see Appendix, Section F), comparing an agent's performance within such a small set can easily lead to biased conclusions. We propose using a broader range of metrics that consider not only the full set of supported tasks but also the size of the dataset used to achieve these performance levels, ensuring a more comprehensive and meaningful evaluation.
>
>
> **3) Updated paper in the pdf format**
>
> We have uploaded the updated version now. We are sorry for the delay.

---

> > ### Author Response · Authors · 2024-12-02
> > **Updated results (Verification of the CAAP prompting with multiple LLMs)**
> >
> > The table below presents the final results of experiments conducted on WebShop to demonstrate the effectiveness of CAAP prompting across different language models. The task scores for all three models range from 61.3 to 68.0, exceeding the score achieved by Pix2Act. These results validate that CAAP prompting offers a robust and generalizable configuration for GUI agent.
> >
> > | Method                   | Image Data Size | Demo Data Size | Task Score |
> > |--------------------------|-----------------|----------------|------------|
> > | **CAAP (Claude Sonnet 3.5)** | 0.3K       | 1              | **68.0**      |
> > | **CAAP (GPT-4)**        | 0.3K            | 1              | 62.3       |
> > | **CAAP (Llama-3.1 70B)** | 0.3K            | 1              | 61.3       |
> > | **Pix2Act**             | 0               | 1K             | 46.7       |
> > | **Human**               | -               | -              | 82.1       |

---

### Official Review · Reviewer_UNBm · 2024-11-03

**Soundness:** 3
**Presentation:** 3
**Contribution:** 2
**Rating:** 5
**Confidence:** 3

**Summary:**

The paper proposes a LLM-agent based framework for computer use that uses only screenshot images as input, converting them to text for LLM processing, operating through basic keyboard and mouse actions on GUI and reducing HTML code and specific APIs. The proposed automation solution CAAP achieves 94.5% success rate on MiniWoB++ benchmark and 62.3 average task score on WebShop as compared to image-only based agents.

**Strengths:**

1. Modular framework: focus on visual observation, decision making and action execution aspects to solve computer tasks. The core components make it suitable for future extensions, maintainability and scalability.
1. Extensive discussion on prompt: for in-context learning from few-shot examples, context info, CoT, extra guidelines. The impact of this prompting scheme is shown as ablation with existing work RCI.
1. Open-source code and models

**Weaknesses:**

1. Architectural Complexity and overhead: seems more complex than end-to-end solutions due to modular design. This also creates specific APIs requirement to connect different modules, fine-tuning for vision models for screenshot specific recognition. It is unclear on what are the implications of these on robustness of the proposed framework.
1. Limited real world evaluation: while the proposed approach is tested on miniwob++ and webshop, it'll be useful to demonstrate it on real world tasks and analyze failure cases.

**Questions:**

1. Is the proposed model specific to certain UI elements in miniwob++ and webshop benchmarks or are they generally applicable?
1. How does the model handle dynamic UI elements (pop-ups), screen resolution, and other visual features when they change position or appearance?
1.  What was the process for selecting the "single human demonstration" used in WebShop instructions? How representative is it? How does it relate to "300 samples collected through a fully automated process"?

---

> ### Author Response · Authors · 2024-11-19
> **(Response to the weaknesses) About architectural complexity and overhead**
>
> The key distinction between our architecture and an end-to-end structure lies in whether the visual observer and the action planner (decision maker) modules are separated. While this modular approach introduces some complexity, the trade-offs yield significant advantages.
>
> For instance, our approach provides an alternative to addressing a fundamental challenge faced by VLMs: the difficulty in accurately identifying the pixel coordinates of objects on screen images. Additionally, it offers advantages such as modular updates, scalability, and parallelization, which are not achievable with end-to-end architectures. Although integrating the modules requires careful interface design, this challenge was manageable through engineering and poses minimal overhead once established, as the interface rarely requires modification after initial setup.
>
> "Fine-tuning vision models for screenshot-specific recognition" is a necessity even for end-to-end structures. However, end-to-end architectures require simultaneous learning of both visual understanding and action trajectories, which significantly increases dataset size and training time requirements. By comparison, our approach offers greater efficiency. Notably, methods such as CC-Net and Pix2Act rely on millions of data samples for their end-to-end training, underscoring the advantages of our modular design.

---

> ### Author Response · Authors · 2024-11-19
> **(Response to the weaknesses) About limited real world evaluation**
>
> There is no doubt about the importance of evaluating real-world tasks. However, given the necessity of choosing from open benchmarks, we had to opt for simulation environments that approximate real-world scenarios. WebShop, for instance, offers a simulated environment built on 1.18M real products sourced from Amazon, focusing on e-commerce product search—a task commonly encountered in real-world scenarios—making it an excellent benchmark for real-world evaluation.
>
> Naturally, WebShop, like other benchmarks, focuses on limited scenarios. To address this limitation, we leveraged MiniWoB++, which is specifically designed to evaluate fundamental computer tasks. Real-world tasks are often composed of combinations of these fundamental tasks, making MiniWoB++ an essential for our evaluation. To arrive at this selection, we thoroughly reviewed various benchmarks. For example, AgentBench and InterCode were unsuitable as they do not support screenshots as input sources. Mind2Web, WebLINX, PixelHelp, MetaGUI, AITW, and OmniAct lacked simulation environments and only provided datasets of successful cases, making them inadequate for validating our approach. Additionally, more recent benchmarks had not yet been sufficiently validated by the community, rendering them unreliable for our purposes.
>
> Given these considerations, selecting MiniWoB++ and WebShop represented the most viable and robust choice for testing real-world task performance. We hope this context clarifies the rationale behind our decision.

---

> ### Author Response · Authors · 2024-11-19
> **Response to the questions**
>
> __Q1) Is the proposed model specific to certain UI elements in miniwob++ and webshop benchmarks or are they generally applicable?__
>
> → Yes, it is generally applicable. The visual observer was trained on the dataset annotated based on UI element types defined according to HTML5 standards. Therefore, even in environments beyond MiniWoB++ and WebShop, it should have little difficulty understanding screens composed of familiar and commonly structured UI elements.
>
>
> __Q2) How does the model handle dynamic UI elements (pop-ups), screen resolution, and other visual features when they change position or appearance?__
>
> → Our visual observer is designed to effectively handle UI elements whose positions and shapes may vary with each instance. For instance, in MiniWoB++ tasks, the "click-button-sequence" task described in Figure 5(d) of Appendix F (failure case study) involves "ONE" and "TWO" buttons placed in different positions across episodes. Despite this variability, our agent achieves a 92% success rate. In the "drag-circle" task, which involves moving a circle shape with randomly assigned position, size, and color in a specific direction, our agent achieves a 98% success rate. Additionally, in the "login-user-popup" task, where a 'session timeout' popup randomly appears during the login process, our agent successfully resolves the issue 100% of the time.
>
> While we did not explicitly validate the ability to handle diverse screen resolutions within our experimental scope, our tests on MiniWoB++ and WebShop included screenshots with different resolutions. Based on this, we believe that as long as the image model is sufficiently trained, the agent should be capable of handling diverse screen resolutions.
>
>
> __Q3) What was the process for selecting the "single human demonstration" used in WebShop instructions? How representative is it? How does it relate to "300 samples collected through a fully automated process"?__
>
> → The instructions in WebShop can be categorized into those requiring the selection of specific product options and those that do not. We sequentially reviewed the instructions and selected the first one that required detailed option selection. Using a custom-built demo recording tool, we created a human demonstration for the chosen instruction. This demonstration covers most scenarios likely to occur in solving WebShop tasks, making it sufficiently representative for addressing the benchmark.
>
> The 300 samples are not related to this human demonstration. While human demonstrations are text-based descriptions of action trajectories successfully solving WebShop tasks, the 300 samples serve as a fine-tuning dataset to enhance the visual observer's understanding of the WebShop environment. These samples comprise screenshot images paired with descriptions of the UI elements displayed. They were collected through a sequential simulation of trainset instructions utilizing an automated process developed with Selenium.

---

> > ### Comment · Reviewer_UNBm · 2024-11-26
> > **Response to the rebuttal**
> >
> > After reviewing the rebuttal, I have reconsidered my original assessment regarding the contribution and decided to maintain my score.
> >
> > I acknowledge that the modular architecture might be better for limited data as compared to end-to-end, but I disagree that "Although integrating the modules requires careful interface design, this challenge was manageable through engineering and poses minimal overhead once established, as the interface rarely requires modification after initial setup." - the setup was not tested with diverse real-world screenshot scenarios to claim that this modular approach scales and generalizes well.
> >
> > Regarding the choice of "open benchmarks" for experiments: it is helpful to compare with established benchmarks, but choosing not to demonstrate on diverse real world setups and show generalization of the approach significantly limits the claims that can be made here.  "even in environments beyond MiniWoB++ and WebShop, it should have little difficulty understanding screens composed of familiar and commonly structured UI elements." - seems like not so usable claim because real world screenshots look nothing like in the currently chosen sim environments.

---

> ### Author Response · Authors · 2024-11-29
> **Addressing Reviewer Concerns on Scalability and Real-World Application**
>
> We acknowledge the reviewer's concern that our modular approach may not scale or generalize effectively to real-world scenarios, necessitating more rigorous testing to demonstrate its true value. We understand that "real-world problem-solving" can mean different things to different audiences. It seems the reviewer is particularly concerned about solving highly generalized problems that pertain to a broad spectrum of computer users worldwide. However, our focus is on addressing the specific, economically valuable needs of enterprises seeking work automation, which also qualifies as solving real-world scenarios.
>
> Enterprises frequently deploy Robotic Process Automation (RPA) systems, which are highly effective in handling everyday, repetitive tasks, thereby enhancing operational efficiency. These RPA systems already incorporate image-processing capabilities that successfully identify and describe UI elements. (It should be noted that these RPA systems are only required to cover a certain small set of computer environments.) The agent design proposed in our paper can seamlessly integrate with existing RPA workflows, enabling task execution driven by LLM-powered reasoning and natural language-based task descriptions, rather than relying on manually crafted rules.
>
> Thus, we respectfully disagree with the claim that further demonstrations are necessary to validate our approach's value. While the complexity of real-world problems, such as interpreting real-world screenshots, may be challenging to analytically decompose, this aspect is already well-addressed by existing RPA systems. Our contribution lies in enhancing these systems with LLM-driven capabilities, offering a clear and practical path for enterprise adoption.

---

### Official Review · Reviewer_tUvm · 2024-11-04

**Soundness:** 3
**Presentation:** 3
**Contribution:** 3
**Rating:** 5
**Confidence:** 3

**Summary:**

This paper introduces CAAP, which is a novel framework for automating complex computer tasks using LLMs by interacting solely through the front-end UI. Unlike previous approaches that rely on HTML source code or application-specific APIs, CAAP perceives its environment through screenshot images, which are then converted into text using a visual observer module (YOLO and Pix2Struct). This text is processed by an LLM with CAAP, which includes human demonstrations, surrounding context information, CoT-inducing instructions, and lastly some extra guidelines. The agent executes keyboard and mouse operations to interact with the GUI, eliminating the need for pre-defined APIs or large-scale human demonstration data. Experiments on the MiniWoB++ and WebShop benchmarks demonstrate that CAAP outperforms existing methods.

**Strengths:**

This method makes a strong push for an agent that interacts with computer tasks using only screenshot images and standard input devices (keyboard and mouse), which more closely mimics human interaction than methods that use HTML or APIs. The overall paper is fairly clear and there is a good suite of baseline experiments and decent ablations. The originality of investigating prompting structure for agentic interactions with standrad input devices is good and is an important direction for future research.

**Weaknesses:**

The main novelty seems to be the CAAP prompting because converting the screenshots to text is done with fine-tuned YOLO and Pix2Struct models. Although Figure 4 is good at ablating different components of CAAP, it would be beneficial to include more analysis of these components and what traits help the prompting. Since the novelty is predicated on this prompting strategy, I believe a more thorough answer to the question of the best-prompting strategy for agents interfacing with just screenshots is interesting and needed. It would also help if Figures 1 and 2 would be more clear using the captions. Also, it would be good to discuss the limitations of the method.

**Questions:**

What is the reason for converting screenshots to text for LLM and not using a VLM as there is other visual information that isn't captured?
What is the reason for the choice of MiniWoB++?
Why fine-tune the visual observer (YOLO + Pix2Struct) for this particular dataset when you could have auxiliary losses for VLM?
Why have an intermediate representation and not have a more end-to-end approach?
Why is having a VLM be an action planner preferable to other longer-horizon policy learning methods that use semantics as part of their input observation?

---

> ### Author Response · Authors · 2024-11-15
> **Our approach to addressing VLM challenges**
>
> The exclusion of "screenshot-to-text" from our contributions, as highlighted in the weaknesses, seems to arise from a misunderstanding of our work when viewed solely through the lens of module-level components. Our primary contribution is the proposal of a novel architecture for GUI agents. To date, there have been no significant attempts to decompose VLM architectures into modular, function-specific components for GUI agent applications. Through this architectural proposal, we address a well-documented challenge faced by VLMs such as GPT-4V and Claude 3.5 Sonnet: the difficulty in locating pixel coordinates of objects within images. By leveraging the YOLO model, known for its robust object detection performance, we achieve precise pixel coordinate detection of GUI elements. Meanwhile, other functionalities are handled by utilizing LLMs and Pix2Struct, offering an alternative solution to this challenge.
>
> In this architecture, the performance of screenshot-to-text has a significant impact on the action planning capabilities of LLMs. Beyond the foundational effort of fine-tuning the visual observer, we enhanced screenshot-to-text performance by defining UI element attributes aligned with HTML5 standards to accommodate diverse task types. Additionally, we applied a novel masking technique (see Figure 2) to the input images for Pix2Struct. Therefore, we assert that it is appropriate to regard screenshot-to-text as an integral part of our contribution within the modular architecture. This inclusion underscores the comprehensiveness of our approach.

---

> ### Author Response · Authors · 2024-11-15
> **Analysis of what traits help the CAAP prompting**
>
> The effectiveness of the Chain-of-Thought (CoT) mechanism lies in its ability to connect pieces of context information and progressively strengthen their cohesion, enriching the reasoning process. CAAP prompting focuses on enabling this mechanism to operate effectively during the action planning process.
>
> Inspired by human action planning, we identified four essential types of surrounding context information necessary for determining the next actions (see Figure 3 for detailed content). These context elements are conveyed to the LLM along with CoT-inducing instructions, which explicitly guide the model to establish semantic connections between them. Using CoT-inducing instructions, the LLM examines the actions taken so far to uncover the relationship between the given task and the action trajectory. It then infers the contextual meaning of these actions and the updated state of the visual environment. Subsequently, referencing the candidates of action types, it refines the subsequent action plan considering the current situation and outputs actionable guidance in a structured text format. In addition to CoT-inducing instructions, we introduced implicit guidance by incorporating rationales into human demonstrations, enabling the LLM to better link context information. Examples illustrating this mechanism in practice are provided in Figures 8 and 9 of the Appendix H.
>
> Figure 4 presents experimental results on how these traits influence task performance. CoT-inducing instructions consist of multiple ingredients, but they must be treated as a complete set to hold sufficient significance. Therefore, their individual impact on performance was not addressed in the ablation study. As shown in the figure, the addition of CoT-inducing instructions improves task success rates by 5.3% (from 79.2% to 84.5%) in the absence of human demonstrations and by 2.5% (from 92.0% to 94.5%) when human demonstrations are provided. Furthermore, augmenting human demonstrations with rationales results in a 3.5% increase in task success rates, from 91.0% to 94.5%. These findings demonstrate that CAAP prompting, by guiding the LLM to first establish relationships among the given context information prior to decision-making, significantly enhances success rates in solving computer tasks.
>
> We will revise the manuscript to ensure that the aforementioned content is adequately incorporated.

---

> ### Author Response · Authors · 2024-11-15
> **Response to the questions**
>
> __Q1) What is the reason for converting screenshots to text for LLM and not using a VLM as there is other visual information that isn't captured?__
>
> → We determined that the image interpretation capabilities of current VLMs are insufficient for effectively solving computer-based tasks. To address this limitation, we adopted a screen-to-text procedure to deliver precisely interpreted information, thereby leveraging the full language comprehension capabilities of the LLM. This approach constitutes a significant aspect of our contribution, with its effectiveness validated through experimental results.
>
> In tasks involving computer interaction, for instance, it is crucial to accurately extract the pixel coordinates of objects on the screen with which the agent needs to interact. This precision is necessary to ensure accurate execution of actions like clicks within the environment. Although state-of-the-art VLMs exhibit strong general interpretative capabilities, they encounter challenges in pinpointing pixel coordinates, and training them sufficiently for this task would require an extensive amount of annotated data. As an alternative, we recognized the need to separate the role of identifying bounding boxes from the responsibilities of the VLM/LLM. To achieve this, we introduced a visual observer, composed of UI element (object) detection and understanding components, which interprets screenshot data into a well-structured text format for the LLM. This approach demonstrated effective performance with significantly less training data.
>
>
> __Q2) What is the reason for the choice of MiniWoB++?__
>
> → We selected MiniWoB++ because it is particularly well-suited to simulate a wide range of fundamental scenarios that may occur in computer tasks.
>
> MiniWoB++ comprises over a hundred of fundamental tasks, covering most basic unit tasks that can arise during computer operations. While other benchmarks, such as WebShop, Mind2Web, WebArena, AITW, and so on, address more realistic scenarios, they often focus on a limited number of complex tasks built in a combination of a small set of fundamental unit actions. Thus, we aimed to demonstrate our agent’s ability to handle diverse situations through MiniWoB++ validation while also proving its capability to address realistic, complex tasks using WebShop.
>
>
> __Q3) Why fine-tune the visual observer (YOLO + Pix2Struct) for this particular dataset when you could have auxiliary losses for VLM? Why have an intermediate representation and not have a more end-to-end approach?__
>
> → As previously mentioned, we determined that a modular architecture offers greater advantages compared to a VLM-based architecture. It is because separating the modules for visual observation and action planning 1) requires significantly less training data, 2) provides structural flexibility, and 3) allows us to utilize highly generalized LLM models with superior performance, such as GPT.
>
>
> __Q4) Why is having a VLM be an action planner preferable to other longer-horizon policy learning methods that use semantics as part of their input observation?__
> → Policy learning requires a substantial number of human demonstration samples and enables effective solutions for a limited range of tasks.
>
> In contrast, we designed an agent that operates with a dramatically smaller number of human demo samples by utilizing an LLM as an action planner. Rather than addressing a restricted set of problems through policy learning, we believe that leveraging the general planning capabilities of LLMs/VLMs is the correct long-term approach and is essential for successfully tackling a wide variety of tasks in a generalized manner. We aim to verify this through our future research.

---

> ### Author Response · Authors · 2024-11-15
> **Caption updates for figures**
>
> We will revise the captions for Figures 1 and 2, as well as Figures 8 and 9, as follows:
>
> __- Caption for Figure 1:__
> "The architecture and task-solving flow of the CAAP agent. When a task is instructed, the agent interprets a screenshot captured in the computer environment through the visual observer. The action decider leverages the reasoning capabilities of the LLM to determine the next actions to take based on the observed state. Once actions are decided, the action executer applies the corresponding keyboard and mouse actions to the environment via the OS interface. This sequence of processes across the three modules continues until the task is completed."
>
> __- Caption for Figure 2:__
> "Comparison of the masking methods for the original Pix2Struct and our UI element understanding model, and an example of the extracted features for our vision observer. While the original Pix2Struct outputs text in a HTML-like format, our model is finetuned to return JSON-style text. (a) An example of the masking style of the original Pix2Struct that covers the target element with an X-box. (b) An example of our semi-masking approach that highlights the target element by outlining it and darkening its surrounding. An example of the extracted features from the image of (b) by our visual observer."
>
> __- Caption of Figure 8:__
> "An example of the CAAP prompt. The raionales contained in the expert demonstrations and the CoT-inducing instructions, which are detailed in the ‘### Instructions ###’ section, help an LLM gradually develop the semantic connectivity between the task description, the action history, and the visual state of the screen, in a CoT manner."
>
> __- Caption of Figure 9:__
> "An example of an LLM response to the CAAP prompt in Figure 8, showing how the LLM improves evidence for action-decision by connecting the surrounding context information."

---

### Author Response · Authors · 2024-11-29
**Summary of Revisions**

**Dear Reviewers,**

We sincerely appreciate your insightful feedback and constructive suggestions, which have significantly improved the quality of our submission.

In light of the discussions with the reviewers, we have revised our manuscript to incorporate our refined thoughts. The changes are highlighted in blue in the updated PDF. Below is a summary of the revisions for your reference:

### **[Major Revisions]**
- To strengthen our argument, we have added a new paragraph in the introduction discussing the challenges associated with existing VLM-based agents.
- We have enhanced the explanation of how the Chain of Thought mechanism is expressed within the CAAP prompting.
- To demonstrate that CAAP prompting performs well regardless of the LLM used, we have included intermediate results using "Claude 3.5 sonnet" as the action proposer LLM model. The final results will be reflected in the published version of the paper.

### **[Minor Revisions]**
- Renamed the "decision maker" module to "action proposer."
- Replaced instances of "LMM" with "VLM."
- Reduced descriptions of the derived effects of modularization to emphasize its practical impacts.
- Revised the captions of Figures 1, 2, 6, and 7 to provide clearer explanations.
- Adjusted the order of the Appendix content to align with the sequence of references in the main text.
- Polished unclear expressions throughout the text.

Thank you for your time and consideration. We look forward to your feedback on the revised manuscript.

Sincerely,

The Authors

---

### Meta-Review · Area_Chair_nHfr · 2024-12-20

**Metareview:**

The paper proposes a modular LLM-based agent for solving GUI-based tasks using screenshots. The work addresses an important problem and demonstrates impressive engineering efforts. The proposed method achieves good performance on two benchmarks.

However, reviewers raised significant concerns:

- While the modular design and the prompting method are interesting, the reviewers found the contributions insufficiently explained or lacking broader generalization. The CAAP prompting was perceived as tailored toward the specific application and not rigorously validated across diverse situations.

- The focus on MiniWoB++ and WebShop benchmarks without real-world scenarios undermines claims of generalizability. The lack of empirical results demonstrating scalability and applicability to various environments was a common critique.

- Reviewers noted unclear explanations of architectural designs and figures. More detailed discussions on prompting strategies and limitations were requested.

While the paper is promising, it needs to further improve the rigor and clarity. Therefore, I recommend rejection.

**Additional Comments On Reviewer Discussion:**

The authors have addressed some concerns in their rebuttal and revised the manuscript. They presented results using additional language models, demonstrating CAAP's effectiveness across architectures. However, the updates could not fully alleviate concerns regarding scientific depth and real-world applicability.

---

### Decision · Program_Chairs · 2025-01-22

Reject